# Drug screening with human SMN2 reporter identifies SMN protein stabilizers to correct SMA pathology

Yiran Wang[1], Chongchong Xu[2,3], Lin Ma[1,4,5], Yongchao Mou[2,3], Bowen Zhang[1], Shanshan Zhou[1], Yue Tian[1], Jessica Trinh[2], Xiaoqing Zhang[1,4,5,6,7,8] ⓘ, Xue-Jun Li[2,3] ⓘ

**Spinal muscular atrophy (SMA), the leading genetic cause of infant mortality, is caused by reduced levels of functional survival motor neuron (SMN) protein. To identify therapeutic agents for SMA, we established a versatile SMN2-GFP reporter line by targeting the human *SMN2* gene. We then screened a compound library and identified Z-FA-FMK as a potent candidate. Z-FA-FMK, a cysteine protease inhibitor, increased functional SMN through inhibiting the protease-mediated degradation of both full-length and exon 7–deleted forms of SMN. Further studies reveal that CAPN1, CAPN7, CTSB, and CTSL mediate the degradation of SMN proteins, providing novel targets for SMA. Notably, Z-FA-FMK mitigated mitochondriopathy and neuropathy in SMA patient–derived motor neurons and showed protective effects in SMA animal model after intracerebroventricular injection. E64d, another cysteine protease inhibitor which can pass through the blood–brain barrier, showed even more potent therapeutic effects after subcutaneous delivery to SMA mice. Taken together, we have successfully established a human *SMN2* reporter for future drug discovery and identified the potential therapeutic value of cysteine protease inhibitors in treating SMA via stabilizing SMN proteins.**

## Introduction

Spinal muscular atrophy (SMA), the leading genetic cause of infant mortality, is characterized by the specific degeneration of spinal motor neurons caused by the mutation in the *SMN1* gene (Pearn et al, 1978; Pearn, 1980; Burglen et al, 1995; Lefebvre et al, 1995). Humans are unique in that they have two *SMN* genes, *SMN1* and *SMN2* (Lorson et al, 1999; Monani et al, 1999; Rochette et al, 2001). The *SMN1* gene generates full-length transcripts (SMN-FL) and

functional survival motor neuron (SMN) protein. Although the sequence of the *SMN2* gene is very similar to that of *SMN1*, a difference in 5 bp in the *SMN2* gene results in disrupted splicing of exon 7, where both SMN-FL transcripts (10%) and transcripts lacking exon 7 (SMN-Δ7) (90%) are generated (Lorson et al, 1999; Monani et al, 1999). In SMA patients, more than 95% of cases have homozygous deletion of the *SMN1* gene; even if patients have *SMN2*, the majority of the gene products are SMN-Δ7 which is non-stable and degrades quickly, and thus could not compensate for the loss of *SMN1* (Pearn, 1980; Lefebvre et al, 1997). There is an inverse correlation between the number of *SMN2* copies and disease severity, suggesting a dose-dependent effect of functional SMN produced by the *SMN2* gene (Lefebvre et al, 1997). Therefore, increasing the functional SMN by targeting the alternative *SMN2* gene in SMA patients has been a promising therapeutic strategy for SMA (Wirth et al, 2006; Zhou et al, 2012; Singh et al, 2013; Cherry et al, 2014; Howell et al, 2014; d'Ydewalle et al, 2017). Although human *SMN2* gene reporters have been generated to screen for candidate drugs, most of these studies used mini-genes that do not have the full expression, splicing, or regulatory elements (Andreassi et al, 2001; Zhang et al, 2001; Lunn et al, 2004; Morse et al, 2012). Building human *SMN2* reporters that contain all the elements of the human *SMN2* gene would be essential to identify therapeutic agents that can effectively rescue motor neuron degeneration in SMA patients.

SMN protein is a ubiquitously expressed protein; however, reduced levels of this protein results in specific degeneration of spinal motor neurons. It has been shown that the splicing of *SMN2* to SMN-FL transcript is much less efficient in spinal motor neurons compared with other type of neurons (Ruggiu et al, 2011). Moreover, the stability of SMN proteins is also affected in SMA. It has been shown that in the presence of low levels of SMN-FL, the oligomerizations of SMN-FL/SMN-FL and SMN-FL/SMN-Δ7 are significantly reduced (Burnett et al, 2009). The reduced levels of SMN have a profound effect on spinal motor neurons, leading to axonal

[1]Brain and Spinal Cord Innovative Research Center, Tongji Hospital, Tongji University School of Medicine, Shanghai, China    [2]Department of Biomedical Sciences, University of Illinois College of Medicine Rockford, Rockford, IL, USA    [3]Department of Bioengineering, University of Illinois at Chicago, Chicago, IL, USA    [4]Key Laboratory of Reconstruction and Regeneration of Spine and Spinal Cord Injury, Ministry of Education, Shanghai, China    [5]Key Laboratory of Neuroregeneration of Shanghai Universities, Tongji University, School of Medicine, Shanghai, China    [6]Tsingtao Advanced Research Institute, Tongji University, Shanghai, China    [7]Shanghai Institute of Stem Cell Research and Clinical Translation, Shanghai, China    [8]Translational Medical Center for Stem Cell Therapy, Shanghai East Hospital, Tongji University School of Medicine, Shanghai, China

Correspondence: xqzhang@tongji.edu.cn; xjli23@uic.edu
Yiran Wang, Chongchong Xu, and Lin Ma are co-first authors

defects and neuronal degeneration. In addition to impaired axonal outgrowth and transport, a recent study reported the dysfunction of mitochondria in mouse NSC-34 cells with SMN knockdown, suggesting that SMN is also important for mitochondrial function (Acsadi et al, 2009). Impaired mitochondrial function (Berger et al, 2003; Ripolone et al, 2015) and increased oxidative stress (Hayashi et al, 2002) have also been reported in SMA patients. We have successfully established human pluripotent stem cell–based models of SMA (Wang et al, 2013b; Xu et al, 2016), which recapitulate disease-specific axonal phenotypes and selective motor neuron degeneration. Interestingly, we also observed mitochondrial defects in SMA spinal motor neurons (Wang et al, 2013b; Xu et al, 2016). These data suggest that axonal defects and mitochondrial dysfunction are important pathological changes in SMA, which serve as unique counter assays to validate candidate compounds.

Taking advantage of recently developed gene targeting technology based on the CRISPR/Cas9 system (Wang et al, 2013a; Mali et al, 2013; Hou et al, 2014; Niu et al, 2014), in this study, we seek to precisely and specifically target the human *SMN2* gene to build a unique SMN2-GFP reporter line for screening drugs for SMA. Upon the successful establishment of the SMN2-GFP reporter in HEK293 cells, we then screened a small compound library (980 compounds) and identified a novel small molecule that can increase the functional SMN. The compound selected from this small library can significantly increase the expression of SMN protein more than 70% in patient cells, emphasizing the usefulness of this human *SMN2-*reporter–based strategy for drug screening. To dissect the mechanism of action of this compound, we further examined the effects of this compound on transcription, protein stability, and ubiquitin proteasome and cysteine protease pathway. Finally, to determine

the role of candidate compounds on motor neuron degeneration in SMA, we examined the effects of the compound on axonal and mitochondrial defects, as well as motor neuron loss using both SMA induced pluripotent stem cells (iPSCs) and SMA mouse models.

# Results

## Successful generation of a versatile human SMN2-GFP reporter line in HEK293 cells for drug screening

To establish a versatile SMN2-GFP reporter, we targeted the human *SMN2* gene in situ using CRISPR/Cas9–mediated homologous recombination (HR) (Wang et al, 2013a; Mali et al, 2013; Hou et al, 2014; Niu et al, 2014) in HEK293 cells. A gRNA that specifically targets the *SMN2* gene was designed (Fig 1A). We then verified the targeting efficiency of this gRNA in HEK293 cells (Fig 1B). Using this gRNA together with the donor plasmid and Cas9, we next transfected HEK293 cells through the calcium phosphate precipitation method. As shown in Fig 1C, gRNA/Cas9 induced double-strand break (DSB) and promoted HR-based DNA repair in the presence of the recombination donor. After HR, GFP was specifically integrated into the *SMN2* gene to generate the SMN2-GFP reporter (Fig 1C). Please note that GFP with TAA stop codon is specifically integrated and replaced the stop codon within exon 8. During pre-mRNA splicing, open reading frame will include the GFP cassette if exon 7 is excluded. In contrast, the TAA stop codon in exon 7 will make the open reading frame lack GFP if exon 7 is included. If a compound could increase SMN-FL splicing efficiency of *SMN2*, the GFP signal will decrease. However, if a compound potentiates *SMN2* transcription,

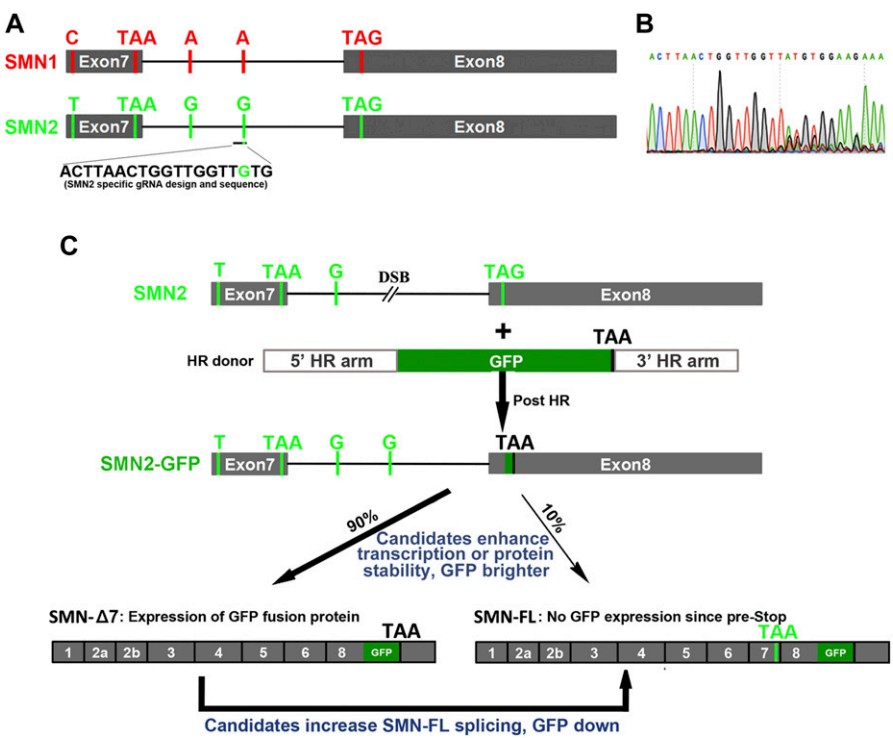

**Figure 1. Building an SMN2-GFP reporter line for drug screening.**
**(A)** Schematic representation of gRNA design targeting *SMN2*. **(B)** gRNA targeting efficiency was verified in HEK293 cells. Overlapped peaks represented NHEJ repair after gRNA/Cas9 induced DSB. **(C)** Schematic strategy for SMN2-GFP reporter line establishment and small molecule screening. gRNA/Cas9 induced DSB promotes HR-based DNA repair in the presence of recombination donor. GFP is specifically integrated into the exon 8 and replaces the stop codon of exon 8. Successful candidate small molecules can either brighten (by transcriptional enhancement or protein stabilization) or darken GFP fluorescence (by splicing pattern switch) in reporter cells, which will be counted as hits for further evaluation.

the GFP signal will increase. Because SMN-Δ7 protein is unstable and degrades quickly, if the SMN protein stability is enhanced by a compound, the GFP signals will also increase. Therefore, the SMN2-GFP reporter we designed is versatile to find candidate drugs, which may regulate *SMN2* transcription, splicing, or protein stability.

Next, the transfected HEK293 cells were dissociated to single cells and cultured clonally. GFP-positive clones were then selected and expanded (Fig 2A). We validated the integration of GFP in these clones which showed correct integration into the *SMN2*-exon 8 as indicated by Southern blot (Fig 2B) and genomic DNA PCR (Fig 2C). As expected, the expression of SMNΔ7-GFP fusion proteins were observed in the reporter lines (Fig 2D). Notably, this GFP integration is specific to the *SMN2* gene as revealed by DNA sequencing (Fig 2E) and does not affect the normal splicing pattern of the human *SMN2* gene (Fig 2F). Together, these data reveal the successful establishment of the versatile human SMN2-GFP reporter line in HEK293 cells.

## Drug screening using the SMN2-GFP reporter identifies a compound that significantly increases SMN protein levels

To test the feasibility of using the human *SMN2* gene reporter for drug screening, we used reported positive compounds for regulating *SMN2* gene transcriptional activation (protirelin) (Kato et al, 2009; Ohuchi et al, 2016), SMN protein stabilization (MG132 and bortezomib) (Burnett et al, 2009; Kwon et al, 2011), and *SMN2* gene splicing (RG7800) (Naryshkin et al, 2014). The *SMN2* reporter cells were treated with these compounds or vehicle for 2 d, and the fluorescence intensities were examined. As expected, protirelin, MG132, and bortezomib significantly increased GFP intensity. In contrast, RG7800, which can down-regulate the proportion of SMN-Δ7 splicing, significantly decreased GFP intensity in SMN2-GFP reporter cells (Fig S1). The Z′-factor for this assay, an evaluation parameter of the quality of assays, is 0.68, suggesting that this reporter system is suitable for drug screening (Zhang et al, 1999). We then examined a small pool library of 980 compounds. SMN2-GFP reporter cells were dissociated and plated onto 96-well plates (10,000 per well). 12 h after plating, each compound was added for 2 d. The fluorescence intensities from each well were then compared between compound-treated and vehicle (DMSO)-treated groups. This resulted in the identification of 14 hits, which brightened GFP fluorescence in the SMN2-GFP reporter line (Fig 2G). Candidate compounds were tested again to validate the repeatability, which showed great repeatability (Fig 2H). Interestingly, there were four candidate compounds belonging to protease inhibitors. Western blot analysis further confirmed that several compounds could

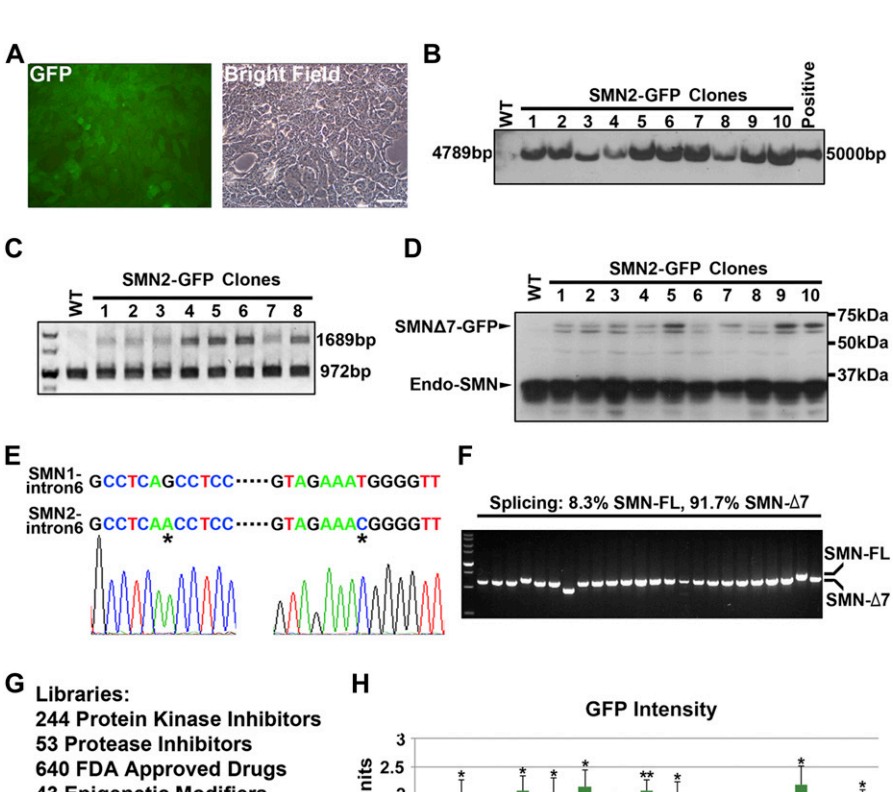

**Figure 2. Establishment of monoclonal SMN2-GFP reporter line in HEK293.**
**(A)** Representative images showing the monoclonal SMN2-GFP reporter line. Scale bar, 50 μm. **(B)** Southern blot showing expected GFP integration. All clones generate a 4,789-bp size DNA fragment comprising the GFP cassette after EcoRI and BamHI digestion. **(C)** Genomic DNA PCR analysis identifying correct GFP integration. **(D)** Western blot showing expression of SMNΔ7-GFP fusion proteins in all reporter lines generated. **(E)** Sequencing of the DNA sequences flanking the GFP cassette confirmed *SMN2* gene rather than *SMN1* gene integration. **(F)** GFP integration in *SMN2* exon 8 showed normal splicing patterns as more than 90% SMN-Δ7 were spliced. **(G)** A small pool library screening identified 14 hits which brightened GFP fluorescence in the SMN2-GFP reporter line. Represented pictures before and after compound #8 (Z-FA-FMK) treatment are shown. Scale bar, 50 μm. **(H)** Quantification data showing that 14 hits significantly increased GFP intensity in the SMN2-GFP reporter cell line by more than 0.5-fold. Data were presented as mean ± SEM, n = 3. *$P < 0.05$, **$P < 0.01$, as compared with DMSO group by two-tailed $t$ test.

significantly increase the protein expression of SMN in patient-derived fibroblasts, among which compound #8 (Z-Phe-Ala fluoromethyl ketone, Z-FA-FMK), an irreversible inhibitor of cysteine proteases (Smith et al, 1988; Van Noorden et al, 1988), was most effective (Fig S2).

Next, we examined the dose-dependent effect of Z-FA-FMK on the SMN expression using the reporter cell line. After treatment with different concentrations of Z-FA-FMK, the GFP fluorescence signals were brighter in cells that were treated with higher concentrations of Z-FA-FMK (Fig 3A). The expression of SMNΔ7-GFP fusion protein in these samples was also analyzed by Western blot, which confirmed a dose-dependent effect of Z-FA-FMK (Fig 3B and C). Interestingly, the mRNA expression of SMNΔ7-GFP was not significantly increased in Z-FA-FMK–treated groups compared with control group (Fig 3D), suggesting that Z-FA-FMK, an inhibitor of cysteine proteases (Smith et al, 1988; Van Noorden et al, 1988), may affect SMN stability.

To evaluate this compound in patients' samples, we examined the effect of Z-FA-FMK on the mRNA and protein expression of SMN in fibroblast cells of SMA patients (GM03813 and GM22592, which were initially diagnosed as type I and type II SMA, respectively). Similarly, as described in the Coriell Web site, these patient fibroblast cells possess the SMN2 gene but do not have the SMN1 gene using TaqMan qRT-PCR assay (Gomez-Curet et al, 2007) (Fig S3). The protein expression levels of SMN in fibroblast cells derived from the SMA patient is much lower than those from the normal individual (WT) (Fig 3E). After treatment with different concentrations of Z-FA-FMK (1, 5, 10, 50, and 100 μM) for 2 d, the expression levels of SMN proteins were increased by Z-FA-FMK in a dose-dependent manner (Figs 3F and S4). Statistical analysis of biological triplicate samples indicated that Z-FA-FMK can significantly increase the expression of SMN proteins starting from 10 μM (Fig S4). Similar results were obtained from a second SMA patient's fibroblast cells (Fig S5), confirming that Z-FA-FMK can restore SMN protein levels in patients' cells. Notably, Z-FA-FMK could increase the SMN protein expression over onefold, which is comparable with a highly potent agent recently reported (Naryshkin et al, 2014). Similarly, as seen in the reporter cells, Z-FA-FMK did not

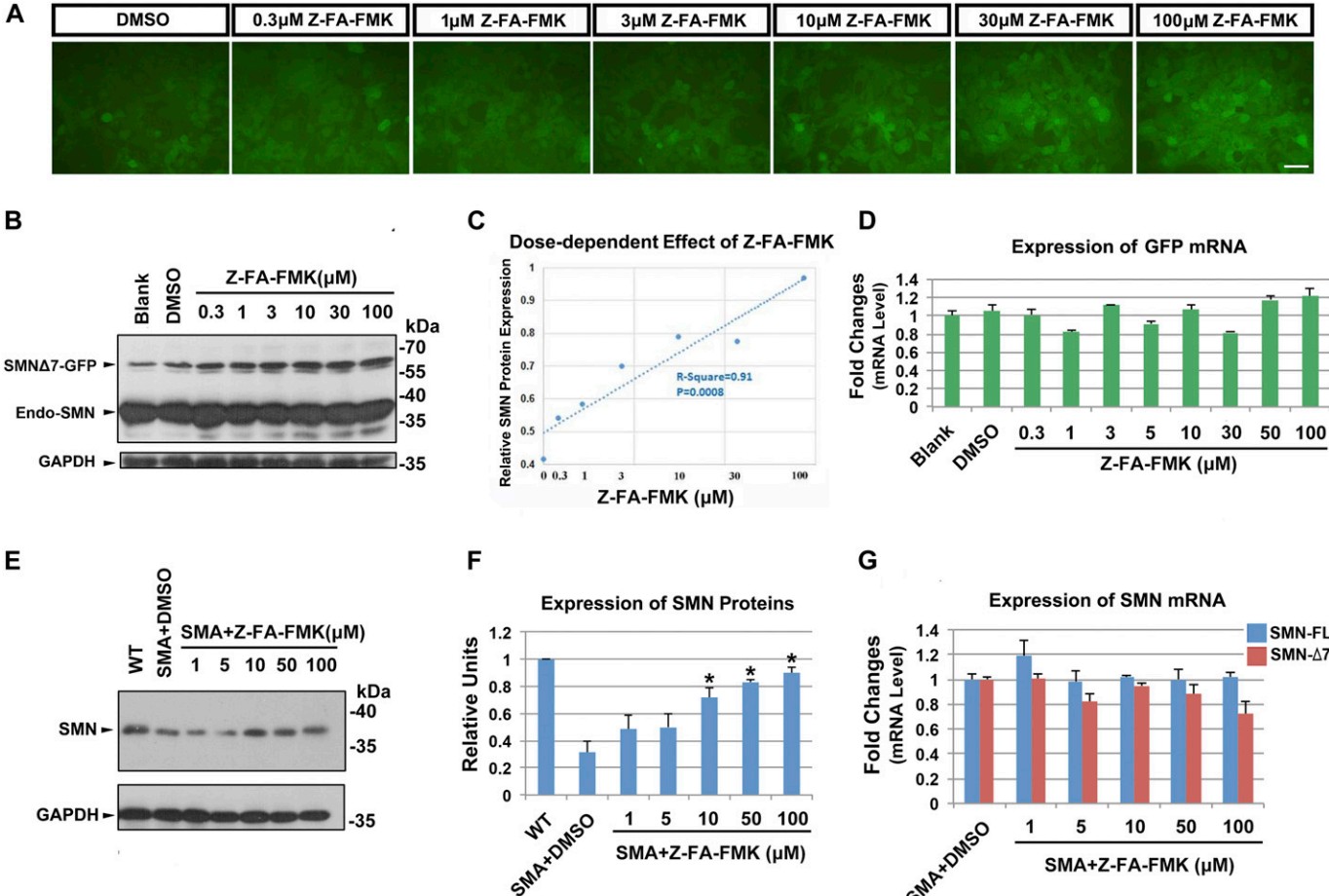

**Figure 3. Identification of Z-FA-FMK as a potent compound that increases SMN protein expression in HEK293 and patient fibroblast cells.**
**(A)** Fluorescent images showed the increased GFP fluorescence at 2 d after treatment with different concentrations of Z-FA-FMK. Scale bar, 50 μm. **(B, C)** Western blot analysis of these samples revealed a dose-dependent increase in the expression of SMNΔ7-GFP proteins after the treatment of Z-FA-FMK. **(D)** qRT-PCR showing the mRNA expression in SMN2-GFP cells after treatment with different concentrations of Z-FA-FMK. n = 3. **(E–G)** In SMA type I fibroblast cells, vehicle (DMSO) or Z-FA-FMK were applied for 2 d, followed by analysis of the protein (E, F) and mRNA (G) expression in these cells. n = 3. Data were presented as mean ± SEM, *P < 0.05, as compared with the DMSO-treated group by two-tailed t test.

significantly enhance the mRNA expression of *SMN* in patient fibroblast cells (Figs 3G and S5C). Together, our data reveal the effectiveness of our *SMN2*-gene–based screening system and identify a small molecule, Z-FA-FMK, which has a novel role in increasing the protein levels of SMN.

### Role of Z-FA-FMK on the SMN expression in SMA patient iPSCs and iPSC-derived motor neurons

To further validate the candidate compound, we performed the secondary analysis using SMA patient–specific iPSCs that were successfully generated from a type I SMA patient (Xu et al, 2016). We first examined the effect of Z-FA-FMK on the expression of SMN proteins in iPSCs (Fig 4A). SMA iPSCs were treated with Z-FA-FMK at different concentrations (1, 10, and 100 μM) for 48 h and subsequently harvested for Western blot analysis (Fig 4B). The protein expression of SMN was significantly increased after treatment with Z-FA-FMK at 10 and 100 μM, revealing a dose-dependent effect (Fig 4B and C). Quantification analyses revealed that the protein level of SMN was significantly increased to ~170% in Z-FA-FMK (10 μM)–treated SMA iPSCs as compared with the DMSO-treated control (Fig 4C).

Given that spinal motor neurons specifically degenerate in SMA patients, we then examined the effect of Z-FA-FMK on the expression of SMN in SMA patient–derived spinal motor neurons. SMA iPSCs were differentiated into spinal motor neurons used a modified monolayer differentiation system in the presence of retinoic acid (for caudalization) and purmorphamine (for ventralization) (Fig 4A). Similarly, as we previously described (Xu et al, 2016), iPSCs first differentiated into neuroepithelial cells (NE, 1-wk) with characteristic rosette structures (Fig 4A). These neural precursors then differentiated into spinal motor neuron progenitors (2-wk) and subsequent post-mitotic spinal motor neurons (3-wk) (Fig 4A) (Xu et al, 2016). Z-FA-FMK was applied to cultures at 10 μM starting from the NE stage (day 7). In day-24 spinal motor neuron cultures, the expression of the functional SMN protein was dramatically increased in Z-FA-FMK–treated group compared with that in DMSO-treated SMA spinal motor neurons (Fig 4D). After further culture, SMA spinal motor neurons degenerated; similarly, as we reported before, there was a significant increase in the caspase 3/7 activity, revealing increased apoptosis caused by the deficiency of functional SMN in long-term cultures. Interestingly, after treatment with Z-FA-FMK, the increased caspase 3/7 activity was significantly ameliorated in SMA motor neuron cultures (Fig 4E). Together, these data suggest that the candidate compound, Z-FA-FMK, is effective in increasing the functional SMN and rescuing motor neuron degeneration in human iPSC-based model of SMA.

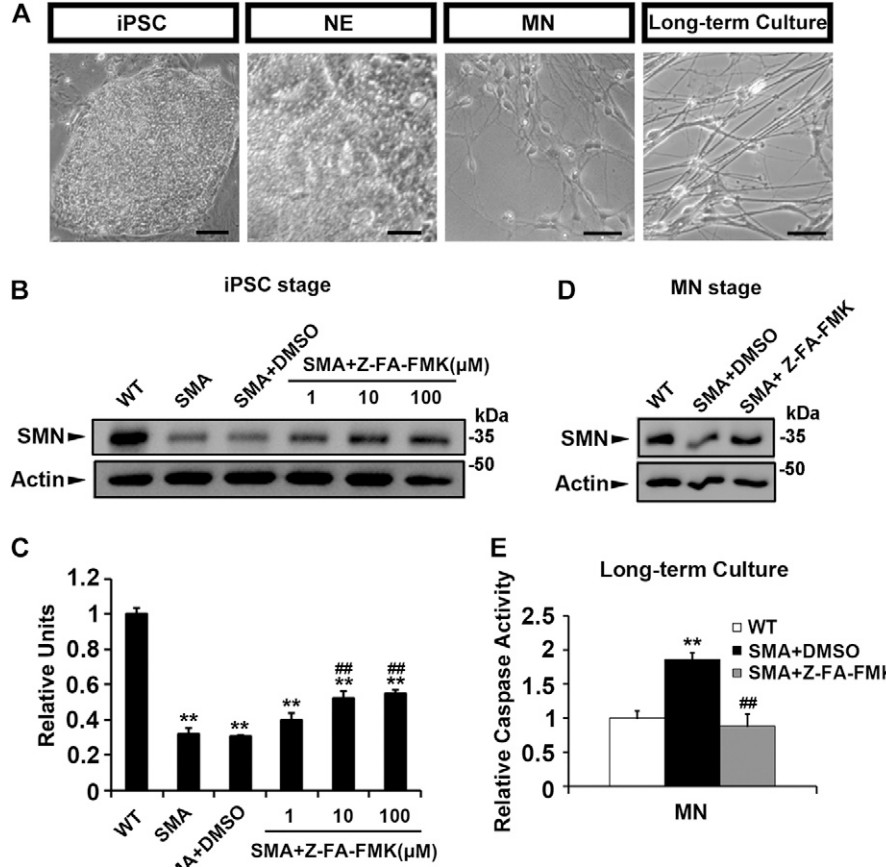

**Figure 4. Effect of Z-FA-FMK on the expression of SMN proteins in SMA patient iPSCs and iPSC-derived motor neurons.**
**(A)** Phase images showing representative stages during motor neuron differentiation from iPSCs (day 0). NE, neuroepithelial cells (1-wk); MN, motor neuron (3-wk). Scale bars, 50 μm. **(B)** Western blot analysis of the SMN protein expression in iPSCs. **(C)** Quantification revealed a dose-dependent increase of SMN proteins in iPSCs derived from a SMA type I patient after Z-FA-FMK treatment. Data were presented as mean ± SEM, n = 3. **(D)** Western blot analysis revealed an increased expression of SMN proteins after Z-FA-FMK treatment at day-24 motor neuron stage. **(E)** The activity of caspase 3/7 was significantly increased in motor neuron long-term cultures (6-wk) derived from SMA iPSCs, and this increase was significantly inhibited by the application of Z-FA-FMK. Mean ± SEM, n = 5. **P < 0.01 as compared with the WT group, ##P < 0.01 as compared with the SMA+ DMSO group by two-tailed *t* test. The dose-dependent effect of Z-FA-FMK **(C)** in SMA iPSCs was confirmed using linear regression (P = 0.0008).

### Effects of Z-FA-FMK on the stabilization of SMN-Δ7 and SMN-FL

The candidate compound we identified from the screening, Z-FA-FMK, is an inhibitor of cysteine proteases (Smith et al, 1988; Van Noorden et al, 1988). To examine the mechanism by which Z-FA-FMK increases SMN levels, we established doxycycline-inducible HEK293 cell lines expressing Myc-SMN2a and Myc-SMN2d to represent different SMN isoforms, SMN-Δ7 and SMN-FL, respectively (Fig 5A). At different time points (2, 4, 8, and 12 h) after doxycycline removal, the expression of Myc-SMN protein was examined and compared between DMSO- and Z-FA-FMK–treated groups. In the DMSO-treated group, both SMN-Δ7 (Myc-SMN2a, Fig 5B and C) and SMN-FL (Myc-SMN2d, Fig 5D and E) proteins were unstable over time, where SMN-Δ7 was noticeably more unstable. Intriguingly, the degradations of both SMN-Δ7 and SMN-FL were significantly inhibited by Z-FA-FMK (Fig 5B–E), revealing that Z-FA-FMK can stabilize both SMN-Δ7

and SMN-FL proteins. It has been shown that MG132, a potent proteasome inhibitor, stabilizes the SMN protein through inhibiting the ubiquitin/proteasome pathway (Chang et al, 2004). Indeed, MG132 led to dramatic increases in both SMN-FL and SMN-Δ7 proteins (Fig 5B–E). To dissect the role of Z-FA-FMK, we performed co-immunoprecipitation and examined the ubiquitination of SMN proteins after applying Z-FA-FMK. Our data showed that Z-FA-FMK, unlike MG132, did not cause the accumulation of ubiquitinated SMN proteins (Fig 5F). Together, these data suggest that Z-FA-FMK has a novel role in stabilizing both SMN-FL and SMN-Δ7 proteins through inhibiting other degradation pathway (e.g., the protease-mediated degradation).

It is interesting that both SMN-FL and SMN-Δ7 can be stabilized by Z-FA-FMK. Next, we examined if the stabilized SMN by Z-FA-FMK is functional. SMN proteins form a complex with Geminis 2–8 and are often localized in Cajal bodies in nuclei, which play an important

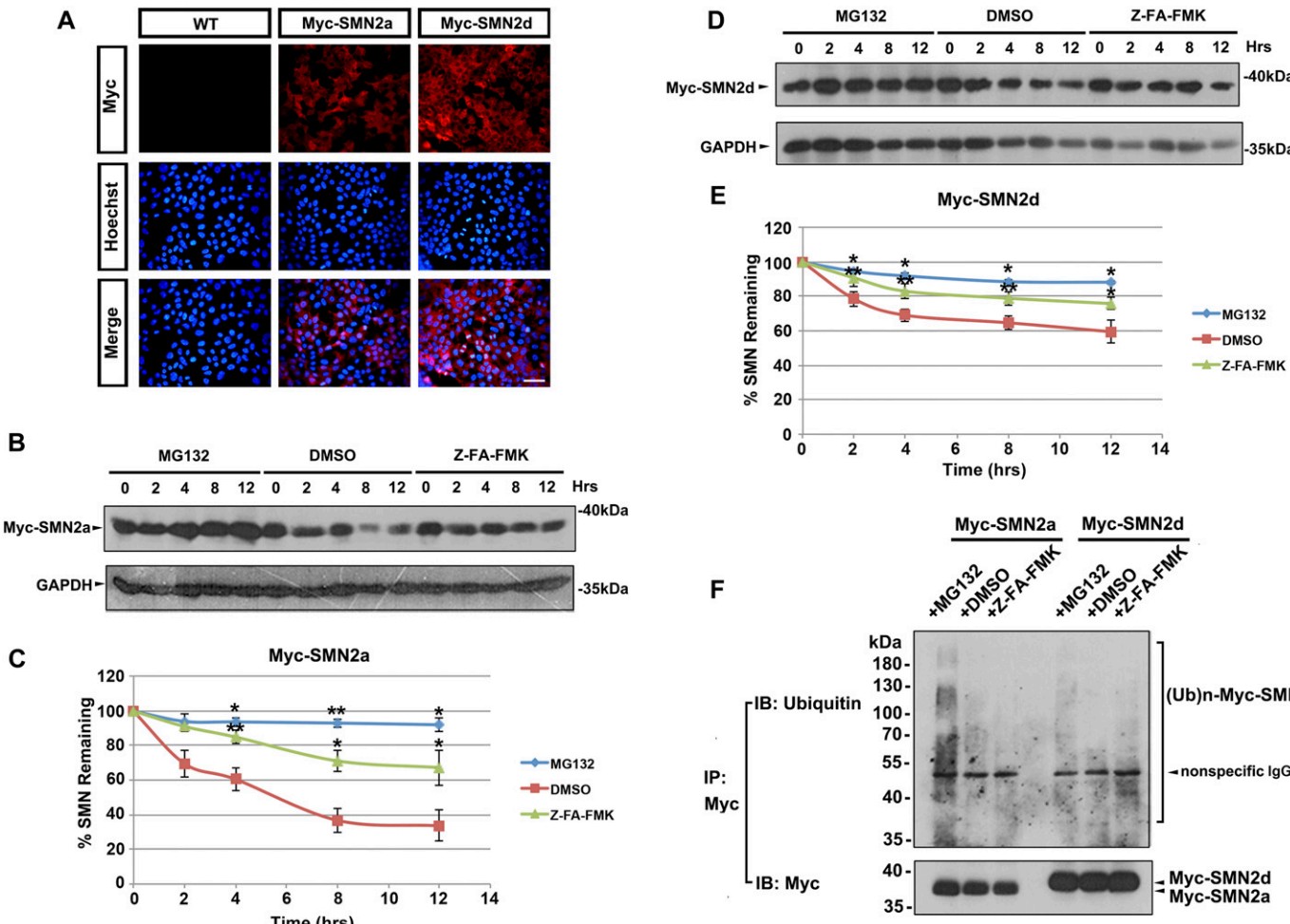

**Figure 5.  Z-FA-FMK stabilized both SMN-FL and SMN-Δ7 proteins in Myc-SMN2a– and Myc-SMN2d–inducible lines.**
**(A)** In Myc-SMN2a– and Myc-SMN2d–inducible lines, addition of doxycycline induced a robust expression of Myc-SMN proteins. Scale bar, 50 μm. **(B–E)** After the withdrawal of doxycycline, inducible cells were treated with Z-FA-FMK, DMSO (vehicle control), and MG132 (positive control) for 12 h to examine the protein degeneration. **(B, D)** Western blot showing the protein expression of the Myc-SMN2a (B) and Myc-SMN2d (D) at different time points after doxycycline withdrawal in Myc-SMN2a– and Myc-SMN2d–inducible cells, respectively. **(C, E)** Quantification from independent experiments showing a significant increase in the expression of both Myc-SMN2a (C) and Myc-SMN2d (E) proteins in the Z-FA-FMK–treated (n = 7 and 5) or MG132-treated (n = 3) group compared with DMSO-treated (n = 7 and 5) group. Data were presented as mean ± SEM. *$P < 0.05$, **$P < 0.01$, as compared with the DMSO group by two-tailed $t$ test. **(F)** Western blot with anti-ubiquitin antibody after co-immunoprecipitation of Myc-SMN showing the ubiquitinated Myc-SMN in the MG132-treated group, rather than in the DMSO- or Z-FA-FMK–treated group.

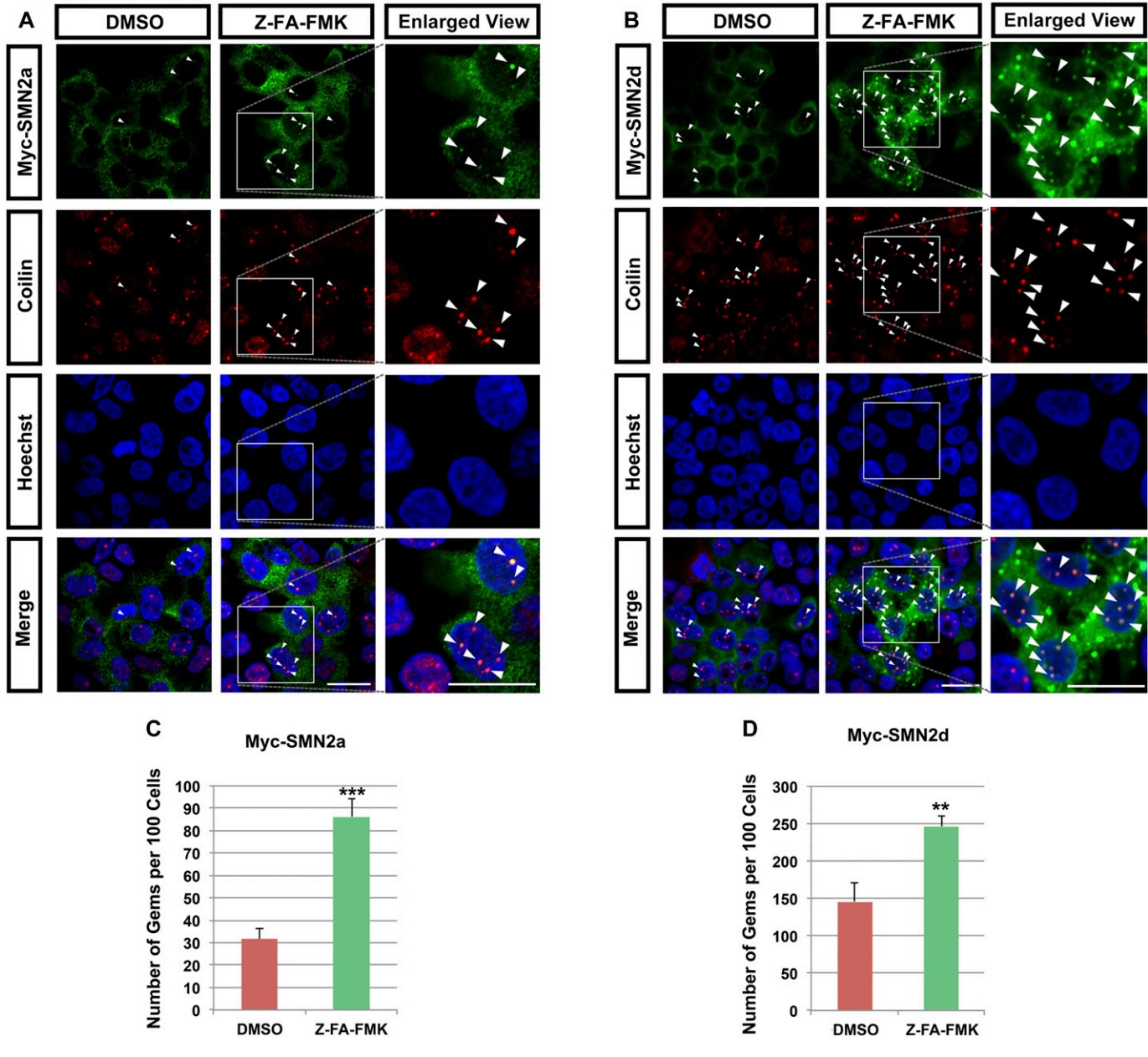

**Figure 6. Both SMN-FL and SMN-Δ7 proteins stabilized by the Z-FA-FMK treatment could form gem-like structures in Cajal bodies.**
**(A, B)** Representative immunostaining images illustrating the expression of Myc-SMN2a and Myc-SMN2d proteins in the DMSO- or Z-FA-FMK–treated groups at 8 h after doxycycline withdrawal. Nuclear Myc-SMN2a/d formed gems in Cajal bodies as indicated by double staining with Coilin (arrows). The third column in these panels indicated the enlarged view of the white-square–labeled area. Scale bars, 20 μm. **(C, D)** Quantification showing a significant increase in the number of both Myc-SMN2a (C) and Myc-SMN2d (D) gems after Z-FA-FMK treatment. Data were presented as mean ± SEM for Myc-SMN2a (n = 4) and Myc-SMN2d (n = 3). **$P < 0.01$, ***$P < 0.001$, as compared with the DMSO group by two-tailed $t$ test.

role in pre-mRNA splicing and gene transcription (Liu & Dreyfuss, 1996; Meister et al, 2002). Using Myc-SMN2a (generating SMN-Δ7 proteins) and Myc-SMN2d (generating SMN-FL proteins)–inducible lines, we compared the formation of SMN gems in Cajal bodies after the withdrawal of doxycycline between DMSO– and Z-FA-FMK–treated groups (Fig 6A–D). As expected, Z-FA-FMK significantly increased the amount of nuclear SMN gems in Cajal bodies after doxycycline withdrawal in Myc-SMN2d line (Fig 6B and D). Notably, the number of gems also significantly increased in Myc-SMN2a cells

after the treatment of Z-FA-FMK (Fig 6A and C). These results demonstrate that both SMN-FL and SMN-Δ7 stabilized by Z-FA-FMK are biologically functional.

### Novel role of cysteine protease-mediated pathway in the degradation of SMN proteins

Z-FA-FMK is reported as an irreversible cysteine protease inhibitor (Smith et al, 1988; Van Noorden et al, 1988), which prompts us to

investigate the role of the protease-mediated degradation on the stability of SMN proteins. Notably, two families of cysteine proteases, calpains (CAPN) and cathepsins (CTS), are reported to be associated with the initiation of protein degradations and neurodegeneration (Yamashima, 2004; Siklos et al, 2015). Calpains are $Ca^{2+}$ ion–activated non-lysosomal cysteine proteases, and cathepsins are lysosomal protease enzymes (Yamashima, 2004; Siklos et al, 2015). Using mRNA sequencing, we first examined the expression of hCalpains and of hCathepsins in 7-wk-old forebrain neurons and spinal motor neurons derived from iPSCs. As shown in Fig 7A, several CAPNs and CTSs, including CAPN1, CAPN2, CAPN7, CTSB, and CTSL are highly expressed (fragments per kilo base per million mapped reads value > 10) in human iPSC-derived motor neurons, implying their role in regulating the degradation of SMN proteins in human motor neurons.

To determine the effects of proteases on the SMN expression, we then screened hCalpains (CANP1-15) and hCathepsins by an RNAi library (Selleck Company) in the established SMN2-GFP reporter cell line. We found through FACS analysis that knockdown of CAPN1, CAPN7, and CTSL could increase SMNΔ7-GFP intensity (Fig 7B and C). Similar results were observed from two different shRNAs that showed effective knockdown of the gene expression (Figs 7C and S6), excluding the off-target effects. We next overexpressed these proteases and found that CAPN1, CAPN7, and CTSB significantly degraded both SMN-FL and SMN-Δ7 proteins (Fig 7D and E). Furthermore, co-immunoprecipitation analyses showed that there was a physical binding between cysteine proteases CAPN1/7 and SMN proteins (Fig 7F). Finally, to investigate the effects of Z-FA-FMK on inhibiting protease-mediated SMN protein degradation, we treated cells that were transfected by combination of proteases (CAPN1+ CAPN7, CTSB+CTSL, and CAPN1+CAPN7+CTSB+CTSL) with Z-FA-FMK for 24 h. These combinations were used to achieve optimal inhibition of SMN proteins for better examining the effect of Z-FA-FMK on both non-lysosomal and lysosomal pathways. Western blot analyses showed that the expression of SMN proteins, especially SMN-Δ7, was significantly reduced after the transfection of these proteases; and this reduction was mitigated, at least partially, by the treatment of Z-FA-FMK (Fig S7). Together, these data demonstrate the novel role of both non-lysosomal (CAPN1 and CAPN7) and lysosomal cysteine proteases (CTSB and CTSL) in mediating SMN protein degradation, and Z-FA-FMK can partially mitigate the degradation. Moreover, these proteases can degrade SMN proteins through either direct binding (e.g., calpain 1/7) or other mechanisms, including lysosomal-mediated pathway (e.g., CTSL/CTSB), providing novel targets for therapeutic interventions for SMA.

### Effects of Z-FA-FMK on the mitochondrial transport defects in SMA

SMA is characterized by axonal and synaptic defects (Evers et al, 1989; Kong et al, 2009; Akten et al, 2011; Dale et al, 2011; Fallini et al, 2013), and our previous study revealed a reduced mitochondrial axonal transport in SMA iPSC–derived spinal motor neurons (Xu et al, 2016). To dissect the protective role of Z-FA-FMK in SMA, we examined the fast axonal transport of mitochondria, which is essential for local synthesis of ATP in areas of axoplasm distant from the cell body. At day 28, the mitochondrial axonal transport was

examined after staining with MitoTracker Red CMXRos and imaged every 5 s for 5 min as we described previously (Denton et al, 2014, 2016) (Fig 8A). After recording and collecting images, we tracked individual mitochondria using MetaMorph image analysis software and quantified various transport parameters (Fig 8B and C). In SMA iPSC–derived spinal motor neuron axons, we observed significant reductions in the percentage of motile mitochondria (Fig 8B) and the frequency of motile events (Fig 8C), revealing the axonal transport defects in SMA spinal motor neurons.

To examine the effects of Z-FA-FMK on the axonal transport defects in SMA, this small molecule was added to SMA iPSC–derived motor neuron cultures at 10 µM starting from the neural progenitor stage (day 7). At day 28, the axonal transport of mitochondria was analyzed in spinal motor neuron cultures and compared with other groups. Although Z-FA-FMK did not show significant effect on the frequency of motile events (Fig 8C), the percentage of motile mitochondria was significantly increased in SMA spinal motor neuron cultures after treatment with Z-FA-FMK (Fig 8B). These data suggest that Z-FA-FMK can rescue, at least partially, the impaired axonal transport of mitochondria in SMA iPSC–derived spinal motor neurons.

### Role of Z-FA-FMK on mitochondrial dynamics and motor neuron loss in SMA

Mitochondria are highly dynamic and constantly undergo fission and fusion to maintain normal shape and function (Chan, 2012; Otera et al, 2013; van der Bliek et al, 2013; Wagener, 2016). In SMA patient iPSC–derived spinal motor neuron cultures, our previous study observed abnormal mitochondrial morphology in these neurons, indicating impaired mitochondrial dynamics. To determine the effect of this small molecule on mitochondrial dynamics, we compared the number and size of the mitochondria between DMSO– and Z-FA-FMK–treated groups (Fig 8D–G). Spinal motor neurons derived from WT control and SMA iPSCs were stained with MitoTracker. After staining, the motor neuron axons were imaged (Fig 8D), and the number and morphology of mitochondria were analyzed using MetaMorph software as we described previously (Xu et al, 2016). In SMA iPSC–derived motor neuron cultures, the number (Fig 8E) and the area (Fig 8G) of mitochondria along axons were significantly decreased compared with that in control neurons. Application of Z-FA-FMK significantly inhibited these decreases, revealing the rescue of mitochondrial dynamics defects.

Spinal motor neurons are large projection neurons that have high energy demand, which specifically degenerate in SMA. Next, we examined if Z-FA-FMK can mitigate the motor neuron loss in long-term cultures (6-wk). Although decreased SMN does not alter the initial specification of motor neurons from human pluripotent stem cells (Wang et al, 2013b; Xu et al, 2016), these neurons degenerate in long-term cultures (Fig 4E). As shown in Fig 9, the proportion of Isl1[+] spinal motor neurons significantly decreased in SMA cultures compared with that in the control group (Fig 9A and B). Notably, the treatment of Z-FA-FMK significantly increased the proportion of spinal motor neurons to a level comparable with control neurons. Moreover, in long-term cultures, axonal degeneration as indicated

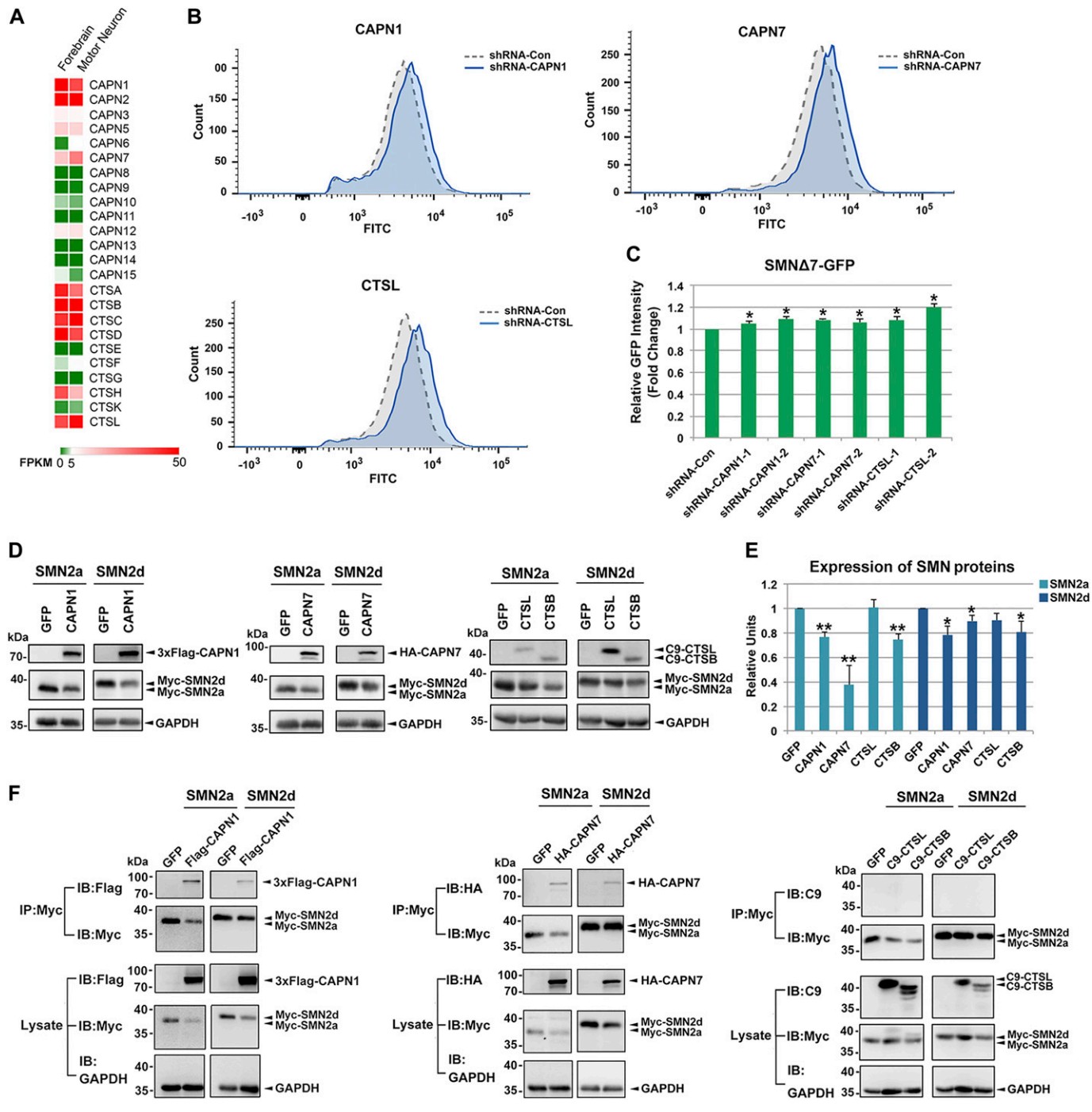

**Figure 7. CAPN1, CAPN7, CTSB, and CTSL mediated the degradation of SMN proteins.**
**(A)** Heat map showing the expression of *hCalpains* and *hCathepsins* in 7-wk-old cultured forebrain neurons and motor neurons derived from normal iPSCs. **(B)** FACS indicating that GFP intensity of SMN2-GFP cells shifted after transfection of *CAPN1*, *CAPN7*, or *CTSL* shRNAs. **(C)** The quantification of SMNΔ7-GFP intensity showing significant increases after knockdown of *CAPN1*, *CAPN7*, or *CTSL*. Data were presented as mean ± SEM, n = 4. *P < 0.05, as compared with the shRNA-Con group by two-tailed *t* test. **(D, E)** Western blot and quantification data showed that overexpression of CAPN1, CAPN7, and CTSB significantly degraded both SMN2a (SMN-Δ7) and SMN2d (SMN-FL) proteins. Mean ± SEM, n = 4. *P < 0.05, **P < 0.01, as compared with the GFP group by two-tailed *t* test. **(F)** Co-immunoprecipitation showing binding between CAPN1/7 and SMN isoforms and no direct binding between CTSB/L and SMN isoforms.

by axonal swelling and breakdown were observed in SMA motor neuron cultures, which was also mitigated by the treatment of Z-FA-FMK (Fig 9C). Thus, our results reveal that Z-FA-FMK effectively

rescues the motor neuron loss in long-term culture in human SMA models, suggesting that Z-FA-FMK or compounds targeting similar pathways serve as potent therapeutic candidates for SMA.

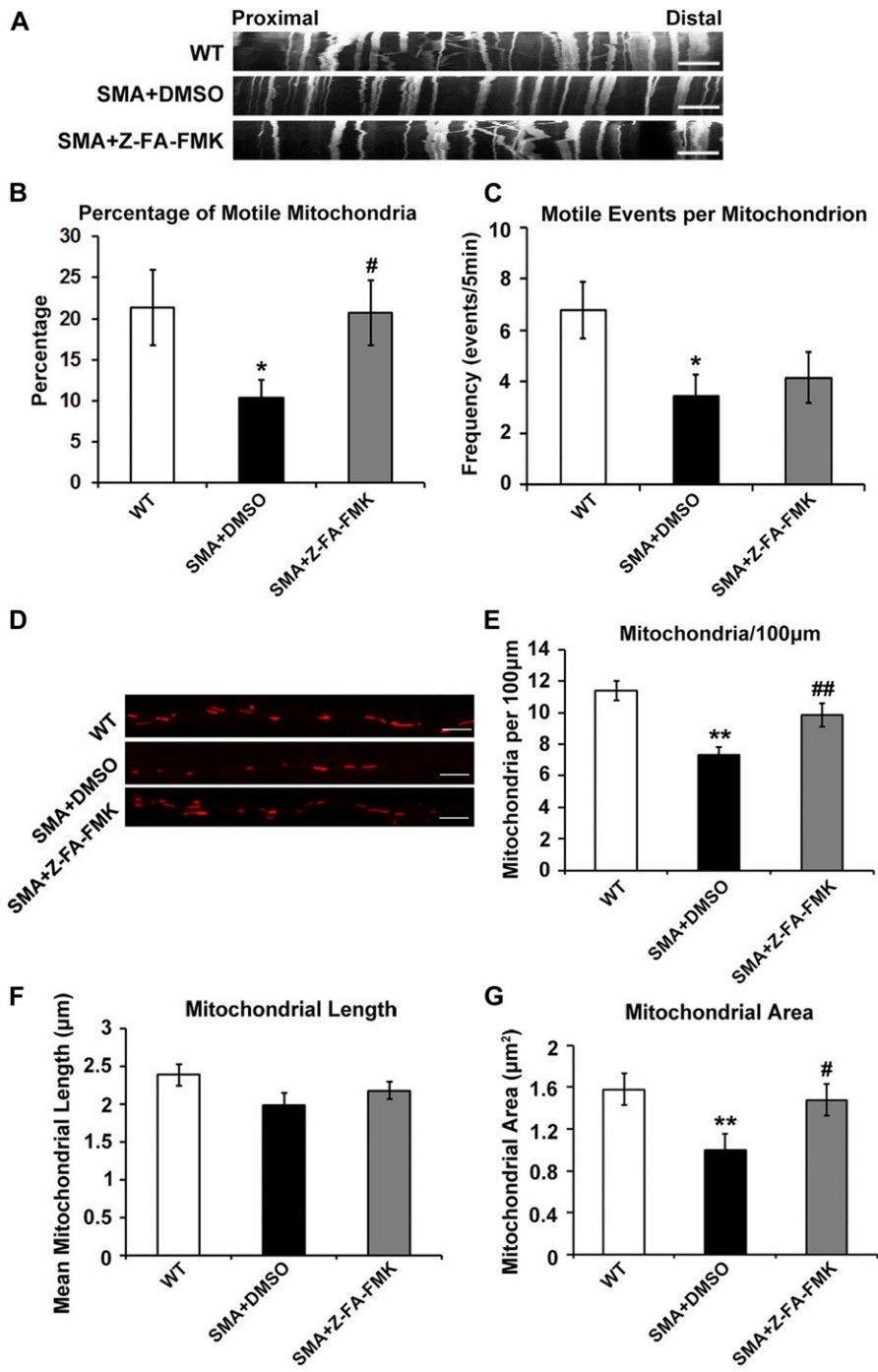

**Figure 8.  Z-FA-FMK rescued the impaired axonal transport of mitochondrial and abnormal mitochondrial morphology in SMA spinal motor neuron cultures.**

**(A)** Representative distance (x-axis) versus time (y-axis) kymographs showing mitochondrial transport in day-28 spinal motor neuron cultures. x-axis in the kymograph represents the positions along neuronal axon and y-axis represents the duration that we detected mitochondrial movement (i.e., 5 min). Scale bars, 10 μm. **(B)** The percentage of motile mitochondria was decreased in SMA iPSC–derived spinal motor neuron cultures. The percentage of motile mitochondria was significantly increased after treatment with Z-FA-FMK. **(C)** The number of motile events per mitochondrion was also significantly decreased in SMA spinal motor neurons. **(D)** Representative images of mitochondria in day-28 spinal motor neuron axons from WT, SMA-, and SMA+Z-FA-FMK–treated groups. Scale bars, 5 μm. **(E)** Average number of mitochondria per 100 μm axon. The number of mitochondria was decreased in SMA iPSC–derived neurons compared with WT group; and this decrease was inhibited by the application of Z-FA-FMK. **(F)** SMA neurons had a nonsignificant trend of decrease in the length of mitochondria as compared with control neurons. **(G)** The mitochondrial area was significantly decreased in SMA neurons compared with WT. After applying the Z-FA-FMK to SMA neural cultures, the mitochondrial area was significantly increased. Data were presented as mean ± SEM for WT (n = 10), SMA+DMSO (n = 13), and SMA+Z-FA-FMK (n = 11). *P < 0.05, **P < 0.01, as compared with the WT group; #P < 0.05, ##P < 0.01, as compared with the SMA+DMSO group by two-tailed t test.

## Protective effects of cysteine protease inhibitor in SMA animal model

To evaluate the role of Z-FA-FMK in in vivo environment, we next tested Z-FA-FMK in SMNΔ7 mouse model. These mice had severe disease phenotypes and died quickly after birth. After we injected SMNΔ7 mice with Z-FA-FMK or vehicle (DMSO) in the lateral cerebral ventricles for only 3 d starting from postnatal day 1 (PND1 to PND3, daily), we found that Z-FA-FMK significantly elevated SMN protein levels in spinal cord at PND4 (Fig 10A) and increased the number of motor neurons in the lumbar spinal cord (Fig 10B), suggesting the protective property of Z-FA-FMK on motor neurons loss through stabilizing SMN proteins in SMA mice. We then investigated the effect of Z-FA-FMK on the life span of SMNΔ7 mice. The mean life span of Z-FA-FMK–treated SMNΔ7 mice was 11.5 ± 1.4 d, whereas control mice survived for 9.4 ± 1.7 d (Fig 10C), revealing a trend of increasing life span after treatment with Z-FA-FMK. There was no significant difference of body weight between groups (Fig 10D).

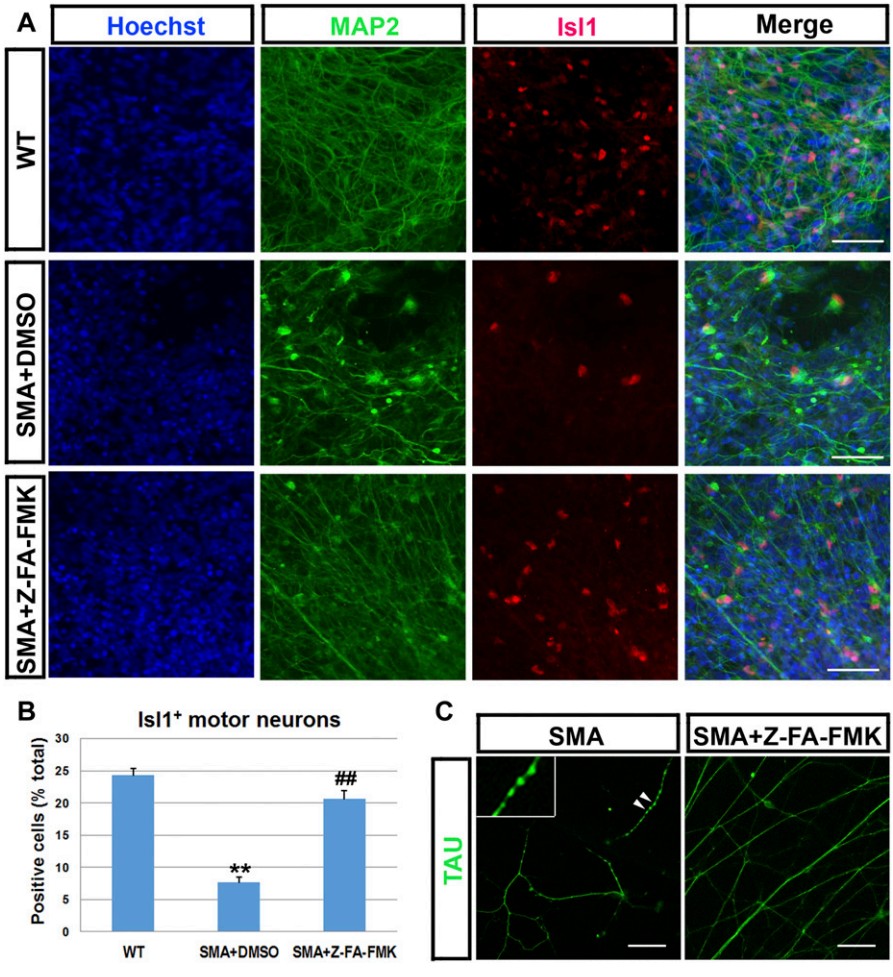

**Figure 9. Z-FA-FMK mitigated the motor neuron loss in SMA long-term cultures.**
**(A)** Immunostaining showing the expression of Isl1+ spinal motor neurons (red) in long-term culture (6-wk) from WT control, SMA+DMSO–treated, and SMA+Z-FA-FMK–treated groups. Green stained MAP2, a neuronal marker, and blue indicated Hoechst-stained nuclei. Scale bars, 50 μm. **(B)** Quantification showing a significant reduction in the proportion of Isl1+ motor neurons in the SMA culture, which can be mitigated by Z-FA-FMK. Data were presented as mean ± SEM, n = 3. **P < 0.01 as compared with the WT group, ##P < 0.01 as compared with the SMA+DMSO group by two-tailed t test. **(C)** Tau immunostaining showing the degenerated axons in the SMA group. Arrowhead pointed area is enlarged in the inset. Scale bars, 50 μm.

Z-FA-FMK has not been proven to cross the blood–brain barrier (BBB); thus, before molecular structure modification, only a narrow treatment window was used, which may be contributed to its relative limited effects on the life span of SMA mice.

Next, to improve the in vivo efficacy, we replaced Z-FA-FMK with (2S, 3S)-trans-epoxysuccinyl-L-leucylamido-3-methylbutane ethyl ester (E64d), another cysteine protease inhibitor which can cross the BBB. Cysteine protease inhibitor E64d is reported to have protective effects for brain injury and Alzheimer's disease (Kuwako et al, 2002; Tsubokawa et al, 2006a, 2006b; Hook et al, 2014) and proved to be safe for use in humans. We first examined the effects of E64d in vitro. We found that E64d, similarly as Z-FA-FMK, also has a trend in inhibiting the degradation of SMN-Δ7 mediated by overexpression of CAPN1, CAPN7, CTSB, and CTSL in HEK293 cells (Fig S7). In the long-term cultures of spinal motor neurons derived from SMA iPSCs, the treatment of E64d significantly increased the proportion of Isl1+ motor neurons, confirming the protective effect of E64d in human SMA motor neuron cultures (Fig 10E and F). These data suggest that E64d is also a potential treatment agent for SMA.

We then tested E64d in SMNΔ7 mouse model after E64d or DMSO subcutaneous injection once daily (starting from PND1). The spinal SMN proteins (Fig 10G) and the number of motor neurons (Fig 10H) in lumbar spinal cord of SMNΔ7 mice were significantly increased by

E64d. Notably, the DMSO-treated mice survived for 8.0 ± 1.0 d. After treatment with E64d, the mean life span of these mice was 12.1 ± 1.1 d, significantly longer (increased 51.2%) than vehicle-treated group using log-rank test (P = 0.016; Fig 10I). There was also an increasing trend in body weight at later time points after E64d treatment (Fig 10J). Taken together, our data demonstrate that the cysteine protease inhibitors, Z-FA-FMK and E64d, can significantly inhibit spinal motor neuron loss and elongate the life span of SMNΔ7 mice, serving as potential therapeutic agents for SMA patients.

## Discussion

SMA is the leading genetic cause of death among infants, and around 50% of SMA patients die before the age of two (Pearn et al, 1978; Prior, 2010; Kolb & Kissel, 2015). Because SMA is caused by reduced levels of functional SMN, to increase the functional SMN by targeting the alternative *SMN2* gene has been a promising therapeutic strategy for SMA (Wirth et al, 2006; Zhou et al, 2012; Singh et al, 2013; Cherry et al, 2014; Howell et al, 2014). A challenge for this approach is to build robust and effective gene reporters that contain all the elements of the human *SMN2* gene. In this study, we

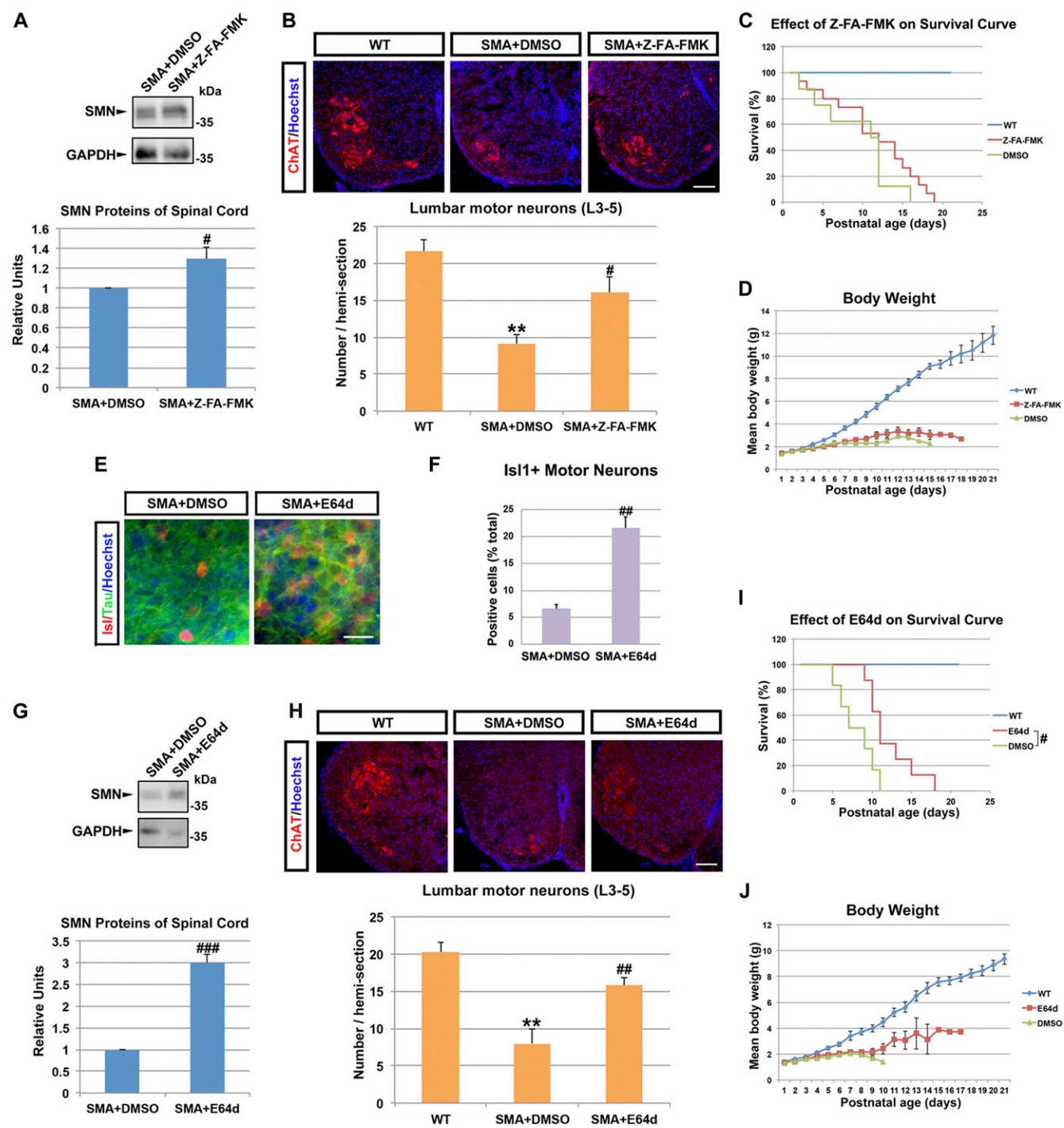

**Figure 10. Protective effects of cysteine protease inhibitor Z-FA-FMK and E64d in SMNΔ7 mice.**
**(A)** Western blot and quantification revealed that spinal SMN proteins of SMNΔ7 mice at PND4 significantly increased after Z-FA-FMK treatment. n = 4. **(B)** Immunostaining and quantification revealed that L3–L5 motor neurons labeled with ChAT antibody in SMNΔ7 mice at PND6 significantly increased after Z-FA-FMK treatment. Scale bar, 100 μm. WT control (n = 6), SMA+DMSO–treated (n = 7), SMA+Z-FA-FMK–treated (n = 5) mice. **(C)** Survival curves of WT control (n = 18), SMA+DMSO–treated (n = 8), and SMA+ Z-FA-FMK–treated (n = 15) mice. **(D)** Body weight of WT control (n = 18), SMA+DMSO–treated (n = 8), and SMA+Z-FA-FMK–treated (n = 15) mice. **(E)** Immunostaining showing the expression of Isl1/Tau in long-term SMA cultures (6-wk) after the treatment of E64d (10 μM) or DMSO. **(F)** Quantification revealed a significant increase in the proportion of Isl1+ spinal motor neurons by E64d; n = 3. **(G)** SMA mice were treated with E64d or DMSO from day 1. Western blot and quantification revealed that spinal SMN protein level of SMNΔ7 mice at PND4 significantly increased after E64d treatment. n = 3. **(H)** Immunostaining and quantification revealed that L3–L5 motor neurons labeled with ChAT antibody in SMNΔ7 mice at PND6 significantly increased after E64d treatment. Scale bar, 100 μm. WT control (n = 6), SMA+DMSO–treated (n = 5), SMA+E64d–treated (n = 5) mice. **(I)** Survival curves of WT control (n = 8), SMA+DMSO–treated (n = 6), and SMA+E64d–treated (n = 8) mice. The statistical significance for the Kaplan-Meier survival analysis was determined by Log-rank test. **(J)** Body weight of WT control (n = 8), SMA+DMSO-treated (n = 6), and SMA+E64d-treated (n = 8) mice. Data were presented as mean ± SEM. **P < 0.01 as compared with the WT mice. #P < 0.05, ##P < 0.01, ###P < 0.001, as compared with the SMA+DMSO group by two-tailed t test.

successfully established a versatile human SMN2-GFP reporter line in HEK293 cells using CRISPR-Cas9–mediated HR. Because we specifically targeted the *SMN2* gene in human cells in situ, an important advantage of this approach is that the *SMN2* reporter has all the regulatory elements and thus will be effective in screening for therapeutic agents. As shown in our study, using the SMN2-GFP reporter, the drug screening of a small pool library effectively identified candidate small molecules that can significantly increase the functional SMN, leading to the rescue of motor neuron degeneration in both the iPSC-based model of SMA and the SMNΔ7 mouse model. Our study also demonstrates that SMN proteins can be degraded by both non-lysosomal and lysosomal cysteine proteases, and inhibitors of cysteine proteases can increase functional SMN through the stabilization of both SMN-FL and SMN-Δ7 proteins, providing novel targets and therapeutic approaches for SMA.

Degeneration of motor neurons in SMA is caused by decreased levels of SMN proteins, and the level of functional SMN in SMA patients is critical for the disease severity and prognosis. Type I will die before the age of two, whereas type II patients who have more copies of *SMN2* could live well into adulthood (Lefebvre et al, 1997). Considering that the functional SMN level in SMA type II patients is ~50% higher than that in type I, more than 50% increase of the functional SMN may have dramatic effects on SMA patients. The increase of functional SMN can be achieved through increasing the transcription, switching the splicing, or stabilizing the SMN proteins. By examining the alterations in the GFP fluorescence intensity in our SMN2-GFP reporter lines, compounds increasing SMN will be able to be identified and will be further validated using patient fibroblast cells and motor neurons. The small molecule (Z-FA-FMK) identified from the present study can increase the expression of SMN proteins, not the mRNA, in SMA patient fibroblast cells and iPSCs in a dose-dependent manner, suggesting that this compound affects the protein stability. Moreover, Z-FA-FMK increased the expression of functional SMN proteins more than 70% in SMA motor neurons, which is comparable with the most potent *SMN2* modifiers (Naryshkin et al, 2014). In addition, cysteine protease inhibitors increase functional SMN through a different mechanism compared with the approved SMA drug (Hache et al, 2016; Bishop et al, 2018), which will allow future combination therapy. Considering the identification of a novel candidate from a small library and the high Z'-factor (>0.5) of the assay, our data suggest the effectiveness of the SMN2-GFP reporter line in identifying potential compounds that can increase functional SMN.

The *SMN2* gene is translated to SMN-FL (~10%) and SMN-Δ7 (~90%) proteins, which can be further degraded through the ubiquitin/proteasome system (Chang et al, 2004; Burnett et al, 2009). SMN-Δ7 is unstable and degrades much quicker than SMN-FL. The half-life of SMN-Δ7 is around half compared with that of SMN-FL (Burnett et al, 2009). Similarly, in our Myc-SMN2a– and Myc-SMN2d–inducible cell lines, the half-life of Myc-SMN2a is around half to that of Myc-SMN2d. Although SMN-Δ7 degrades quickly, a study using severe SMA mice showed that expression of SMN-Δ7 extended the life span of these mice, suggesting that SMN-Δ7 is beneficial in SMA models (Le et al, 2005). One possible mechanism underlying this protective effect is that SMN-Δ7 can form a complex with SMN-FL (Le et al, 2005; Burnett et al, 2009), which is more stable and, thus, enhance the SMN-FL proteins. Our

study demonstrates that the candidate compound Z-FA-FMK can stabilize both SMN-FL and SMN-Δ7 proteins, which contributes to its potent effect in increasing functional SMN proteins. Importantly, both SMN-FL and SMN-Δ7 proteins have the ability to form gems in Cajal bodies in nuclei, suggesting that these stabilized proteins are functional. Furthermore, unlike MG-132, Z-FA-FMK did not cause the accumulation of ubiquitinated SMN proteins, suggesting that Z-FA-FMK increases the SMN stability through a different mechanism other than ubiquitination/proteasome pathway.

Z-FA-FMK is an irreversible inhibitor of cysteine proteases, and a recent study also showed that SMN may be cleaved by calpain 1, a calcium-dependent cysteine protease (Fuentes et al, 2010). However, it is not known whether other cysteine proteases are involved in the degradation of SMN and whether inhibiting cysteine proteases have beneficial effects in SMA models. Using knockdown and overexpression experiments, we identified critical proteases (CAPN1, CAPN7, CTSB, and CTSL) that mediate the degradation of SMN proteins, which serve as novel therapeutic targets for SMA. Interestingly, we observed differential effects after knocking down and overexpressing cysteine proteases. For example, overexpression of CTSL has no effects, whereas knockdown of CTSL increases SMN levels. This may be because that CTSL is important for the initiation of protein degradation, which awaits further investigation. In the knockdown experiments, although the increase of SMNΔ7-GFP by shRNAs that target individual cysteine protease is only 5–20%, the total increase of SMN (both SMN-FL and SMN-Δ7) through inhibiting the activity of multiple proteases would be higher. Similarly, overexpressing the combination of cysteine proteases significantly reduces the SMN protein levels, which can be partially restored by Z-FA-FMK and E64d, suggesting that these compounds can stabilize SMN proteins through inhibiting cysteine proteases. Moreover, compounds that increase SMN stability can be combined with drugs targeting other mechanisms (e.g., increasing SMN expression) in the future for combination therapy. During the review of our article, a study revealed that calpain 1 can degrade SMN proteins, and inhibition of calpain is beneficial for SMA (de la Fuente et al, 2018). This study is coinciding with our findings, and more importantly, our data demonstrate that both non-lysosomal (e.g., calpain 1/2) and lysosomal cysteine proteases (e.g., CTSL/CTSB) are involved in degrading SMN proteins. Lysosomal dysfunction is a critical pathological change in neurodegenerative diseases, and, thus these lysosomal cysteine proteases could have important roles in SMA. The detailed alterations of both non-lysosomal and lysosomal cysteine proteases, as well as their roles in the pathogenesis of SMA, will be investigated in the future.

In SMA iPSC–derived spinal motor neurons, axonal and mitochondrial defects were observed followed by specific degeneration of spinal motor neurons, recapitulating the disease-specific phenotypes. To test the efficacy of candidate compounds, we examined their effects on these phenotypes using both iPSC and animal models. The reduced axonal mitochondrial transport and abnormal mitochondrial morphology, as well as late motor neuron degeneration, can be mitigated by Z-FA-FMK. Notably, application of Z-FA-FMK for only 3 d (PND1 to PND3) was able to increase the life span of SMNΔ7 mice, confirming the efficacy of Z-FA-FMK in vivo. Because Z-FA-FMK could not pass the BBB, this drug was administrated intracerebroventricularly for only 3 d. To increase the time

window of drug administration, we tested E64d, a protease inhibitor, which can pass the BBB. Despite the potential pleiotropic effects by inhibiting proteases, E64d has been shown to be safe to patients during clinical trials for Alzheimer's disease and traumatic brain injury (from *Medtrack*). Our data reveal that E64d significantly increases functional SMN and mitigates motor neuron degeneration in SMA cell models in vitro. The neuronal activity of these SMA motor neurons could be further examined using electrophysiology analysis in the future. Notably, using SMA mice, we found that injection of E64d significantly elevates SMN protein levels, increases the number of spinal motor neurons, and extends the life span of SMA model mice, confirming its efficacy in vivo. Taken together, our study identifies the novel role of Z-FA-FMK and E64d in rescuing motor neuron degeneration in SMA and highlights a potential therapeutic approach through inhibiting protease-mediated degeneration pathway for the treatment of SMA.

# Materials and Methods

### Generation and validation of SMN2-GFP reporter

The *SMN2* targeting donor vector comprised homology sequences and a GFP cassette. 2 $\mu$g targeting donor, 0.5 $\mu$g *SMN2*–specific gRNA, and 0.5 $\mu$g Cas9 plasmid were transiently cotransfected into HEK293 cells (Invitrogen) using the calcium phosphate precipitation method. Both GFP start codon and exon 8 stop codon were omitted after HR. Cells were selected and enriched by GFP fluorescence. For clonal culture, 64 single cells were plated in each 96-well plate. After 10–12 d, positive single clones with GFP fluorescence were picked up and validated by genomic DNA PCR. PCR products were then sequenced to confirm correct integration according to the two unique sequence sites differing between the *SMN1* (NG_008691.1) and *SMN2* (NG_008728.1) genes within intron 6. Primer sets used are as follows: SMN2-5′arm, 5′-ACGTCCTGCAGGTGCC-CAGGGTGGTGTCA-3′, 5′-ACGTCGTCTCATCACTGCCAGCATTTCCTGCAAAT-3′; SMN2-3′arm, 5′-ACGTCGTCTCAAGCAGCACTAAATGACACCA-3′, 5′-ACGTGCGGCCGCTAATTTAAAAAAAAATTAAATATTTTTATTATATACTTTT-3′; GFP, 5′-ACGTCGTCTCAGTGAGCAAGGGCGAGGAGCT-3′, 5′-ACGTCGTCTCATGCTT-TACTTGTACAGCTCGTCCATGC-3′; genomic DNA PCR, Forward 5′-GAGCT-CAGGTGATCCAACTGTC-3′, Reverse, 5′-TTAGTGGTGTGTCATTTAGTGCTGC-3′; Intron 6-F 5′-GAGACAGAGTCTTGCTCTGTCA-3′, GFP-R 5′-CTTCAGGGT-CAGCTTGCCGTA-3′.

To determine the splicing pattern of SMN2-GFP, 1 $\mu$g total RNA of each positive clone was reverse-transcribed into cDNA and removed by RNase H (Invitrogen) in 37°C for 20 min. SMN2-GFP cDNA was amplified by the primers: SMN-exon4-F, 5′-AATCAGATAA-CATCAAGCCCAAATC-3′, GFP-R, 5′-CTTCAGGGTCAGCTTGCCGTA-3′, and subsequently ligated into the pMD19 T-vector (Takara). TA Clones in *Escherichia coli* were assessed by PCR analysis. Different SMN isoforms could be separated on a 2% agarose gel.

### Drug screening in a small library of compounds

SMN2-GFP reporter cells were seeded into 96-well plates at a density of 10,000 cells per well. After 12 h, a small library with protease inhibitors, protein kinase inhibitors, epigenetic modifiers,

and FDA-approved drugs was delivered to cell cultures. Compounds from the library were added to each well individually with a final concentration of 10 $\mu$M. For each treatment plate, DMSO (0.1%) was used as negative control. MG132 (10 $\mu$M final concentration; Selleck), bortezomib (50 $\mu$M final concentration, added at 16 h before analysis; Selleck), protirelin (10 $\mu$M final concentration; Selleck), and RG7800 (1 $\mu$M final concentration; MedChemExpress) were used as positive controls. After 48 h, the change in GFP signal intensity was captured by a fluorescence microscopy (Olympus IX71) and analyzed by ImageJ. Single cells were circled by a constant 10 × 10 $\mu$m frame in ImageJ (Fig S1). GFP intensity in the frame was recorded and then divided by the area of selected region to get average optical density. 100 frames from different fields of each well were calculated to get average GFP optical density of cells. Dead cells were avoided when its density was 0.5-fold brighter than the average density of the entire well (Fig S1). Likewise, the average optical density of background (no cell fields) was calculated. The ultimate GFP fluorescence intensity of cells was equal to average GFP optical density minus average background optical density. Those compounds that could brighten or darken the GFP fluorescence by more than 0.5-fold compared with that of DMSO-treated cells were regarded as potential hits. The Z′ factor, an effective measurement for assessing the quality of screening assays, was determined from the fluorescence intensities of positive samples using ImageJ as described before (Zhang et al, 1999).

### Southern blot

10 $\mu$g genomic DNA was digested with EcoRI and BamHI (NEB), which would generate a 4,789-bp band if GFP was correctly integrated. Digested DNA was electrophoresed and transferred to a nylon membrane (Millipore). A 5,000-bp linearized plasmid bearing a GFP sequence was used as a positive control. A DIG-labeled GFP probe was assembled according to the manufacturer's instructions (Roche). The membrane was hybridized in the hybridization buffer at 55°C (Roche), washed in 2× SSC/0.1% SDS at room temperature and 0.1× SSC/0.1% SDS at 65°C. The hybridized blot was blocked and then probed by anti-DIG antibody (1:10,000; Roche) and afterwards incubated with a fast chemiluminescent substrate (Roche) of alkaline phosphatase for detection. The GFP probe was generated with the following primers: 5′-ACGTAAACGGCCACAAGTTC-3′, 5′-GAACTCCAGCAGGACCATGT-3′.

### Building inducible Myc-SMN2a and SMN2d lines

N-terminal Myc-tagged SMN2a and SMN2d were cloned into the pLVX-Tight-Puro vector (Clontech) through EcoRI and BamHI sites (Zhang et al, 2010). Ef1a-rtTA was cloned into a pLenti vector (Addgene). 10 $\mu$g lentiviral vector, 7.5 $\mu$g PAX2, and 5 $\mu$g VSVG plasmids were cotransfected into HEK293 cells using the calcium phosphate precipitation method in a 10-cm dish. Fresh medium was supplied 16 h post transfection. The medium containing viral particles was collected and filtered after 2 d. HEK293 cells were then incubated with the Myc-SMN2a (or Myc-SMN2d) and rtTA viral supernatants to make inducible lines. At 48 h after viral infection, puromycin was delivered to select for drug-resistant clones.

Myc-SMN2a/d cells were induced by doxycycline for 2 d. Subsequently, the cells were washed thoroughly with DMEM medium for three times and supplied with DMEM/10% KOSR medium for degradative SMN protein experiments.

## Co-immunoprecipitation and ubiquitination assay

Myc-SMN2a/d cells were first induced to overexpress Myc-tagged SMN proteins with doxycycline for 2 d and then washed to abrogate inducement. The cells were treated with 10 μM Z-FA-FMK, 10 μM proteasome inhibitor MG132, or DMSO. After 4 h, the cells were lysed in RIPA buffer with protease inhibitor cocktail (Sigma-Aldrich). Protein concentrations were determined with the BCA kit (Thermo Fisher Scientific). Myc-SMN proteins were subjected to immunoprecipitation with anti-Myc antibody (Cell Signaling Technology)–conjugated G-Agarose (Roche) at 4°C and washed three times with RIPA buffer before being subjected to Western blot analysis.

## Motor neuron differentiation of iPSC lines

Human iPSC lines, including WT control and SMA iPSCs were generated from normal and SMA fibroblast cells (Coriell Cell Repositories) using episomal method as we described before (Xu et al, 2016). These iPSCs were cultured on a feeder layer of irradiated MEFs with the hESC media (+10 ng/ml FGF-2) changed daily. To generate spinal motor neurons from iPSCs, the iPSCs were first differentiated to neuroepithelia in a neural medium in the presence of SB431542 (2 μM), DMH1 (2 μM), and CHIR99021 (3 μM) for 7 d. At day 8, the neuroepithelia were treated with retinoic acid (0.1 μM) and purmorphamine (0.5 μM) for spinal motor neurons induction as we described before (Xu et al, 2016). Spinal motor neurons were efficiently generated at around 3 wk after differentiation (Xu et al, 2016). The expression of various proteases in spinal motor neurons and forebrain neurons (7-wk-old, derived from WT iPSCs using the suspension method) was examined by mRNA-sequencing using an Illumina HiSeq 2000 sequencer. The data were deposited in the ArrayExpress database (accession number E-MTAB-7770).

## Western blot

Cell pellets were collected and lysed in lysis buffer with protease inhibitor cocktail (Sigma-Aldrich). The particulate fraction was removed by centrifugation. Proteins (20–40 μg) were separated on 10% SDS–PAGE and subjected to immunoblotting analysis (Li et al, 2009). Primary antibodies used were rabbit anti-actin (1:1,000; Sigma-Aldrich), rabbit anti-GAPDH (1:2,000; ExCell Bio), mouse anti-SMN (1:1,000; Abnova), mouse anti-Myc (1:2,000; Cell Signaling Technology), rabbit anti-ubiquitin (1:1,000; Cell Signaling Technology), mouse anti-FLAG (1:1,000; Sigma-Aldrich), rabbit anti-HA (1:1,000; Cell Signaling Technology), and mouse anti-rhodopsin (Huang et al, 2006) (1:1,000; Santa Cruz). Horseradish peroxidase-conjugated secondary antibodies were detected with Western Lighting Chemiluminescence Reagent Plus (Pierce). For quantifying SMN proteins, SMN band intensities were normalized with actin or GAPDH and compared between different groups using ImageJ software.

## qRT-PCR

RNA was isolated from cells using TRIzol reagent (Invitrogen) following the manufacturer's instructions. A total of 1 μg of RNA was used to synthesize cDNA using iScript cDNA Synthesis kit (Bio-Rad) according to the supplier's protocol and was used as templates for the quantitative real-time polymerase chain reaction (qRT-PCR). qRT-PCR reactions were performed in a 20-μl mixture containing cDNA, primers, and 1× SYBR GREEN PCR Master mix (Bio-Rad). Expression levels of the mRNA were calculated using the comparative Ct method. The following primers were used in the study: SMN-FL: 5′-GCTGATGCTTTGGGAAG-TATGTTA-3′, 5′-AATGTGAGCACCTTCCTTCTTTTT-3′; SMN-Δ7, 5′-ATTCTCTT-GATGATGCTGATGCT-3′, 5′-TGCCAGCATTTCCATATAATAG-3′; GFP, 5′-ACCACTACCAGCAGAACA-3′, 5′-GAACTCCAGCAGGACCAT-3′; GAPDH, 5′-ATGACATCAAGAAGGTGGTG-3′, 5′-CATACCAGGAAATGAGCTTG-3′; CAPN1, 5′-GAAGCGTCCCACGGAACTG-3′, 5′-GTGCAGGAGGGTGTCGTTG-3′; CAPN7, 5′-TGGCTCGACAGGCACTAGA-3′, 5′-AGGTGGCTTTGGCTTAACACT-3′; CTSL, 5′-AAACTGGGAGGCTTATCTCACT-3′, 5′-GCATAATCCATTAGGCCACCAT-3′.

## Proteases RNAi screening

SMN2-GFP HEK293 cells were transiently transfected with a U6-promotor–triggered shRNAs library (Selleck Company), including *hCalpain* family members, *hCathepsin B, hCathepsin L, hCathepsin S,* and a random sequence as a negative control. The knockdown efficiency was confirmed in normal HEK293 cells. Fresh medium was supplied 12 h post transfection. After 2 d, GFP intensity was decided by FACS analysis; cells transfected with shRNA-Con as negative control was served for GFP gating. The targeting sequences of shRNA-CAPN1-1: AGGAGATTGACGAGAACTT; shRNA-CAPN1-2: CAATTCCTCCAAGACCTAT; shRNA-CAPN7-1: TGCGATTTATCCAGTTGAA; shRNA-CAPN7-2: GGCA-TAATTTACCCTCAAA; shRNA-CTSL-1: AAAGACCGGAGAAACCATT; shRNA-CTSL-2: GGGAGAAGAACATGAAGAT; shRNA-Con: TTCTCCGAACGTGTCACGT.

## Proteases overexpression

HEK293 cells were transiently co-transfected with 0.5 μg 3xFlag-tagged CAPN1 (50941; Addgene), HA-tagged CAPN7, C9-tagged CTSB (11249; Addgene), C9-tagged CTSL (11250; Addgene), or GFP plasmid (negative control), 0.5 μg doxycycline-inducible pLVX-Tight-Myc-SMN2a/d, and 0.5 μg Ef1α-rtTA plasmid. Fresh medium containing doxycycline was supplied 12 h post transfection to induce the expression of Myc-SMN isoforms. After 2 d, the cells were harvested in RIPA buffer with protease inhibitor cocktail (Sigma-Aldrich) for degradation analysis by Western blot. 20 μM Z-FA-FMK or E64d was added to cell cultures for 24 h to study their effects on SMN stabilization. For protein interaction analyses between SMN isoforms and proteases, 1 μg protease plasmid, 1 μg pLVX-Tight-Myc-SMN2a/d, and 1 μg EF1α-rtTA plasmid were co-transfected in HEK293 cells for co-immunoprecipitation.

## Live cell imaging with MitoTracker

Spinal motor neuron progenitors were plated onto polyornithine- and laminin-coated 35-mm dishes (MatTek). At day 28 of total differentiation, the cells were stained with 50 nM MitoTracker Red CMXRos (Invitrogen) for 3 min to allow visualization of

mitochondria. Live-cell imaging was performed using a Carl Zeiss Axiovert microscope equipped with an incubation chamber. Axons identified according to morphological criteria (constant thin diameter, long neurites, no branching, and direct emergence from the cell body) were imaged every 5 s for 5 min, yielding 60 frames. Quantifications were performed using the same protocol as described previously (Denton et al, 2016). In short, the location of each mitochondrion was manually selected using the Track Points function in MetaMorph, and parameters such as the distance from cell body and velocity were recorded. Mitochondria that changed positions in at least three consecutive frames were considered motile.

To analyze mitochondrial morphology, the same straightened images that were generated for measuring mitochondrial transport were used (Denton et al, 2016). We measured the length of each imaged axon and divided it by the number of mitochondria within the region to analyze the mitochondrial density. To analyze the mitochondrial area, the total mitochondrial area was measured and divided by the number of mitochondria within the region.

### Immunocytochemistry and quantification

Coverslips were fixed with 4% paraformaldehyde and immuno-histochemistry was performed as described previously (Li et al, 2005, 2009). Antigen–antibody reactions were developed by appropriate fluorescence-conjugated secondary antibodies. Nuclei were visualized by Hoechst staining. Primary antibodies used in this study included rabbit anti-Myc (1:200; Cell Signaling Technology), mouse anti-Coilin (1:250; Santa Cruz), mouse anti-Islet (1:100; Developmental Studies Hybridoma Bank), rabbit anti-MAP2 (1:500; R&D), rabbit anti-Tau (1:200; Sigma-Aldrich), and goat anti-ChAT (1:200; Millipore). The population of Isl1-expressing motor neurons among total differentiated cells (Hoechst labeled) was counted as described previously (Li et al, 2009). For analyzing the gem numbers, blindly selected fields were imaged from ~100 cells per group. The cells were fixed at 8 h after doxycycline withdrawal. The number of Myc-SMN gems was counted as the punctate foci co-labeled with Coilin, a Cajal body marker protein, and divided by the total number of Myc$^+$ cells in each field. At least three coverslips and three to four fields of each coverslip were chosen and counted.

### Analysis of caspase 3/7 activity

For measurements of the activities of caspase 3 and 7, the Caspase-Glo 3/7 Assay (Promega) was carried out according to the manufacturer's instructions. Briefly, spinal motor neuron cultures were dissociated with Accutase (Invitrogen) and seeded into 96-well plates at 5,000 cells per well in 50 $\mu$l of caspase-3/7 reagent. After incubation for 1 h at room temperature, luminescence from each well was then measured using Wallac Victor2 1420 MultiLabel Counter.

### Animal

All mice experiments were performed following the protocols approved by the Animal Care and Use Committee at the University of Illinois and Tongji University. SMNΔ7 heterozygous mice (*SMNΔ7$^{+/+}$, SMN2$^{+/+}$, Smn1$^{+/-}$*) were purchased from The Jackson Laboratory (005025). Carrier mice were used for breeding, and the offspring were marked and divided into two groups for either compound or vehicle (DMSO) treatment. All mice were genotyped after death or weaning by PCR protocols according to The Jackson Laboratory. 60 ng Z-FA-FMK (dissolved in 5% DMSO) or 5% DMSO diluted by 1 $\mu$l artificial cerebrospinal fluid (Hara et al, 1997) was injected to the lateral cerebral ventricle of mice once daily starting at PND1 until PND3 following the established protocol (Hammond et al, 2017). Neonatal pups were placed on ice for hypothermic anesthesia till their cessation of movement. Lateral ventricles were identified by 2/5 distance from λ suture to eye and 3 mm ventral from skin. A 1-$\mu$l micro syringe with an OD 0.5-mm needle was used. After injection on the right hemisphere, pups were held for warming till recovery of movement and then placed back to the dam cage. E64d (5 mg/kg, dissolved in 1% DMSO diluted by corn oil) or vehicle (1% DMSO in corn oil) were subcutaneously injected once per day after birth. Each mouse was weighed daily from PND1.

To measure SMN protein level in spinal cord, compound-treated, DMSO-treated, or WT samples were collected at PND4 in Eppendorf tubes, weighed, and homogenized in 20 $\mu$l/mg RIPA buffer with protease inhibitor cocktail (Sigma-Aldrich). To assess the motor neuron pathology, the mice were anesthetized at PND6 by Avertin and perfused with PBS followed by 4% paraformaldehyde. Lumbar spinal cord segments three to five were fixed, dehydrated, and transected at 25 $\mu$m. Motor neurons were immunostained with anti-choline acetyltransferase (ChAT) antibody. 6–8 sections of each mice and a total of 5–7 mice from each group were evaluated.

### Statistical analysis

The statistical significance of mean differences among different sample groups was analyzed using two-tailed $t$ test. The linear regression with the log transformation for dose [log(dose + 1)] was used to examine the dose-dependent effect of Z-FA-FMK. For analyzing the life span of SMA mice, the Kaplan–Meier survival analysis with log-rank test was used to compare the differences between groups. The significance level was defined as $P < 0.05$.

# Acknowledgements

This work was supported by the National Key Research and Development Program of China (2018YFA0108000), the Blazer Foundation, the National Institutes of Health (R21NS089042), the National Science Foundation China (31872760, 31400934), the Shanghai Municipal Education Commission (C120114), the Fundamental Research Funds for the Central Universities, and Major Program of Development Fund for Shanghai Zhangjiang National Innovation Demonstration Zone (Stem Cell Strategic Biobank and Clinical Translation Platform of Stem Cell Technology, ZJ2018-ZD-004).

### Supplementary Information

## Author Contributions

Y Wang: data curation, formal analysis, investigation, methodology, and writing—original draft.

C Xu: data curation, formal analysis, investigation, methodology, and writing—original draft.

L Ma: data curation, formal analysis, investigation, methodology, and writing—review and editing.

Y Mou: formal analysis, methodology, and writing—review and editing.

B Zhang: investigation, methodology, and writing—review and editing.

S Zhou: investigation, methodology, and writing—review and editing.

Y Tian: investigation and writing—review and editing.

J Trinh: methodology and writing—review and editing.

X Zhang: conceptualization, formal analysis, supervision, funding acquisition, investigation, project administration, writing—original draft, review, and editing.

X-J Li: conceptualization, formal analysis, supervision, funding acquisition, investigation, writing—original draft, project administration, and writing—review and editing.

## Conflict of Interest Statement

The authors declare that they have no conflict of interest.

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
