## [Reviewer comments · Life Science Alliance]

Life Science Alliance

Drug screening with human SMN2 reporter identifies SMN protein stabilizers to correct SMA pathology

Xue-Jun Li, Yiran Wang, Chongchong Xu, Lin Ma, Yongchao Mou, Bowen Zhang, Shanshan Zhou, Yue Tian, Jessica Trinh, and Xiaoqing Zhang

DOI: <https://doi.org/10.26508/lsa.201800268>

Corresponding author(s): Xue-Jun Li, University of Illinois and Xiaoqing Zhang, Tongji University

Review Timeline:

Submission Date:	2018-12-06
Editorial Decision:	2018-12-21
Revision Received:	2019-02-28
Editorial Decision:	2019-03-05
Revision Received:	2019-03-12
Accepted:	2019-03-12

Scientific Editor: Andrea Leibfried

Transaction Report:

Please note that the manuscript was previously reviewed at another journal and the reports were taken into account in inviting a revision for publication at Life Science Alliance prior to submission to Life Science Alliance.

Referee #1 Review

Remarks for Author:

Wang et al. developed a drug screening system with a human SMN2 reporter cell line and found that a compound increased the SMN2 protein level. The flow presented in this manuscript is straightforward and easy to understand, but unfortunately the novelty is not strong in terms of the screening strategy to find enhancers of SMN2 expression (previously discussed elsewhere, e.g. *Drugs*. 2018 Mar;78(3):293-305., *J Biomol Screen*. 2012 Apr;17(4):481-95. Etc.), and investigation for MOA of lead compounds is limited or not sufficient. Although the approach is interesting, there are several issues to be resolved for scientific significance.

Major points:

1. Evaluation of the HTS assay system is insufficient. Positive control is lacking and information of the HTS method is poor. The authors described that they used fluorescence microscopy, but only the name of the light source was written in the methods without explanation of the HTS system. It is unclear how HTS was conducted, how the fluorescence intensities were evaluated using Image J, and how autofluorescence of dead cells was distinguished.

Furthermore, the result of the screen showed poor reproducibility as presented in Figure 2H. 10 of 14 hit compounds did not show the increase of SMN2 protein by Western blotting. The SMN2 expression pattern in HEK293 might be completely different from that of patient fibroblasts, animal models, and patients, and could have the possibility of abnormal karyotype, including the locus in SMN1 or SMN2, after long-time passages. When the authors use patient fibroblasts to confirm the therapeutic effect of hit compounds, control fibroblasts should be set at the same time.

2. The authors described that Z-FA-FMK could elongate the life span of SMA model mice in the results section. However, Z-FA-FMK did not show any positive effect on mouse survival with statistical significance. Furthermore, it is unclear why E64d, which was not a hit compound, was selected in the next in vivo experiments, even though there were several cysteine protease inhibitors. The authors should present the rationality.

3. Details of the method for in vivo experiments are lacking. The authors should describe how they injected Z-FA-FMK into lateral cerebral ventricles of postnatal day 1 - day 3 mice, and also how they decided the dose of compounds. It is unclear whether the amount of Z-FA-FMK 60 ng (155 micromolar) of 1 microliter per day is appropriate for the treatment.

4. The authors should investigate the SMN2 protein levels in spinal cord of SMN model mice after treatment with Z-FA-FMK and E64d to clarify the POC in vivo.

Minor points

1) The quality of all images to show altered GFP signals is poor with a high background noise. The authors had better use the Luc reporter system.

2) The authors should add graphs to show the screening results by plotting altered ratios compared with DMSO control for each compound.

3) In Figure 2H, Western blotting band is not appropriate. GAPDH band is also altered by the addition of compounds, including compound #8 up to 2-fold change. The authors should also add the calculated data to show the changes.

4) In Figure 4A, characterization of differentiated motor neurons is lacking. The authors should add the information of cell-specific markers and their purity (percentage per DAPI etc.)

5) In Figure 4B, the band of actin is also altered when adding Z-FA-FMK.

Referee #2 Review

Remarks for Author:

Wang et al generated human SMN2-GFP reporter cell line, performed high-throughput screening, and identified candidates that could increase SMN protein level. Especially, they found an irreversible inhibitor of cysteine proteases Z-FA-FMK as a most effective candidate. They further tested the effect of Z-FA-FMK in patient iPSCs and iPSCs-derived motor neurons, and found that it is related to protease/degradation pathway inhibition. Furthermore, authors demonstrated that 3 cysteine proteases CAPN1, CAPN7, and CTSL were important in mediating SMN protein degradation. Cellular phenotypes of SMN, the abnormal mitochondria behaviors and the neuron degeneration in motor neurons, were also rescued by Z-FA-FMK treatment. Finally, they demonstrated that Z-FA-FMK and another cysteine protease inhibitor had protective effects in SMA animal models. Overall, the study was well organized and clearly presented, and the conclusions were strongly supported by their results. Minor comments include the below:

1. For most of the quantification data shown, the author indicated that mean {plus minus} s.e.m. was shown. The information about replicate/n number for those quantifications should be provided.
2. In Figure 4A, the time point for each stage should also be clearly stated either in the figure, or in the legend.
3. In Figure 7, the authors identified 3 cysteine proteases CAPN1, CAPN7, CTSL as important to regulate SMN protein stability. However, the effects of Z-FA-FMK on these specific proteases were not examined or discussed.
4. In Figure 8A, this figure is difficult to understand. The author should clearly explain the results.
- 5., In Figure 8 and 9, mitochondria transport and neuron degeneration were examined to confirm the functional relevance of SMN protein stabilization mediated by Z-FA-FMK. Ideally, neuronal activities should be examined to eventually confirm the functional relevance, or the author should discuss about this.
6. In Figure 9A and B, although there was a decrease in ISI1+ motor neurons in SMA+DMSO group, the proportion of DAPI+ nuclei looks similar (i.e. similar numbers of total cells were still growing in each condition). How do the authors address the effects of differentiation variations/motor neuron purity among different groups in their quantification? Similar issue is for Figure 10C and D.
9. In Figure 9C, the author may also enlarge part of the figure to better display the axon swelling and breakdown.
- 10, In Figure 10C and D, there is no WT control. Besides, if there is no DAPI staining for total nuclei, how did the author quantify the percentage of positive cells (Figure 10D)?

Referee #3 Review

Remarks for Author:

This manuscript, "Drug screening with human SMN2 reporter identifies SMN protein stabilizers to correct SMA pathology", describes construction of SMN2 reporter cell line and its application to screen for compounds that correct SMN2 exon7 skipping, a common cause of SMA. A modest scale screen identified a compound Z-FA-FMK, a cysteine protease inhibitor, which increases the amount of SMN Δ 7. Z-FA-FMK had a positive effect on SMN's stability and on several phenotypic measures, including in SMA patient iPSC-derived motor neurons, as well as a small increase in survival of SMA mice model. The manuscript includes several interesting observations, however, it does not in my view add up to a significant advance and questions remain unaddressed.

Several concerns:

The susceptibility of SMN Δ 7 to proteases has been previously elucidated and the potential of inhibitors of this process for SMA have been previously proposed by studies from Burnett/ Fishbeck, Dreyfuss, and Rubin laboratories.

Protease inhibitors, such as Z-FA-FMK, are likely to have pleiotropic effects rather than selective effect on SMN Δ 7. This needs to be investigated systematically, for example, by mass spectrometry. It is possible, and whatever effects Z-FA-FMK has may be unrelated to the SMN increase and will be deleterious to cells or an organism in longer term experiments.

Comment: Similar SMN2-based reporter screens for SMA have been previously described, but the authors seem to have a nice set up which may be useful for larger scale screens.

December 21, 2018

Re: Life Science Alliance manuscript #LSA-2018-00268-T

Dr. Xue-Jun Li
University of Illinois
Department of Biomedical Sciences
1601 Parkview Avenue
College of Medicine at Rockford
Rockford, IL 61107

Dear Dr. Li,

Thank you for transferring your manuscript entitled "Drug screening with human SMN2 reporter identifies SMN protein stabilizers to correct SMA pathology" to Life Science Alliance. The manuscript was assessed by expert reviewers at another journal before, and the editors transferred those reports to us with your permission. Furthermore, you had already provided an outline on how you could address the main concerns of the reviewers.

The reviewers thought that your work was well-executed, but noted that similar approaches have been used before and that insight into how SMN stabilization occurs is lacking. This is not a concern for publication here, and we would thus like to invite you to provide a revised version of your manuscript. Please provide a point-by-point response to all concerns raised and include the information and additional data that you already outlined upfront / provided upfront for editorial assessment. Please also make sure to adequately describe the HTS method (reviewer #1) and to add the requested quantifications. The potential pleiotropic effects of protease inhibitors should furthermore get mentioned in the manuscript text.

Thank you for this interesting contribution to Life Science Alliance. We are looking forward to receiving your revised manuscript.

Sincerely,

Andrea Leibfried, PhD
Executive Editor
Life Science Alliance
Meyerhofstr. 1
69117 Heidelberg, Germany

t +49 6221 8891 502
e a.leibfried@life-science-alliance.org
www.life-science-alliance.org

- A letter addressing the reviewers' comments point by point.
- An editable version of the final text (.DOC or .DOCX) is needed for copyediting (no PDFs).
- High-resolution figure, supplementary figure and video files uploaded as individual files: See our detailed guidelines for preparing your production-ready images, <http://life-science-alliance.org/authorguide>
- Summary blurb (enter in submission system): A short text summarizing in a single sentence the study (max. 200 characters including spaces). This text is used in conjunction with the titles of papers, hence should be informative and complementary to the title and running title. It should describe the context and significance of the findings for a general readership; it should be written in the present tense and refer to the work in the third person. Author names should not be mentioned.

B. MANUSCRIPT ORGANIZATION AND FORMATTING:

Full guidelines are available on our Instructions for Authors page, <http://life-science-alliance.org/authorguide>

Major concerns from reviewers:

1. The susceptibility of SMN proteins to proteases has been previously elucidated.

Though previous studies have examined the stability of SMN proteins, they have been focused on ubiquitin proteasome and Calpain 1 (summarized in following **Table 1**). Instead, **we have examined two major families of cysteine proteinases and studied their effects systematically from multiple angles**. We first performed a screen for **these proteases** using RNAi (**Fig. 7**), then evaluated the candidates using co-Immunoprecipitation and Western Blot to exam their binding with SMN isoforms (**Fig. 7**), and finally examined the degradation of SMN proteins after overexpression of cysteine proteases using inducible Myc-SMN2a and SMN2d (**Fig. 7**; **Fig. S7**). Our data showed that both non-lysosomal (i.e. Calpain 1/7) and lysosomal cysteine proteases (e.g. CTSL/CTSB) can degrade SMN-full length and SMN- $\Delta 7$. These data are significant because except Calpain 1, the role of Calpain2/CTSL/CTSB in degrading SMN proteins have not been reported. Moreover, cathepsins (CTSB/CTSL) belong to lysosomal cysteine protease and lysosome dysfunction has been implicated in many neurological diseases. **Thus, we demonstrate that SMN proteins can be degraded by both non-lysosomal and lysosomal cysteine proteases (through either direct binding or other mechanisms including lysosomal-mediated pathway)**, providing novel targets for regulating SMN proteins.

Table 1: Summary of previous studies on SMN protein stability (before our submission)

	Molecules	Principle	Reference
1	Proteasome inhibitor	SMN is ubiquitinated and degraded by the ubiquitin proteasome system.	(Abera et al., 2016; Burnett et al., 2009)
2	Gene edition	SMN $\Delta 7$ splicing defect creates a potent degradation signal (degron) at SMN $\Delta 7$'s C-terminal 15 amino acids.	(S and G, 2010)
3	G418, TC007	SMN C-terminus modulates protein stability in a sequence-independent manner that can be corrected by translational readthrough.	(Ebert et al., 2009; Heier and DiDonato, 2009; Wolstencroft et al., 2005)
4	GSK-3 inhibitor	SMN is phosphorylated by GSK-3 and degraded much more rapid.	(Makhortova et al., 2011)

5	Autophagy inhibitor	SMN is degraded through p62-dependent autophagy.	(Rodriguezmuela et al., 2018)
6	LDN-75654 and its analog	Screened for libraries and identified candidates to increase SMN protein expressions; compounds showed protective effects in SMA models. The mechanisms, however, were not clear; neither through inhibiting proteasome nor through inhibiting autophagy.	(Cherry et al., 2013; Rietz et al., 2017)
7	ALLN	This work showed that SMN is a proteolytic target of Calpain 1. ALLN was used a tool to inhibit Calpain 1 but its effect was not tested in any SMA cell or animal models. Exogenous Calpain 1 was also used.	(Fuentes et al., 2010)

2. Effects of Z-FA-FMK and E64d on SMN levels and motor neuron degeneration *in vivo* in SMA mice were not clear

We performed additional experiments to examine these and our data revealed a significant increase in the numbers of ventral horn motor neurons in lumbar segments after Z-FA-FMK and E64d treatment, which is coinciding with the significant increase of SMN proteins in spinal cord tissues after treatment (**Fig. 10**). This is a significant advantage over previous study about Calpain 1 and SMN (Fuentes et al., 2010), in which the effects of inhibiting cysteine proteases were not studied in any cell or animal SMA models. Our new data confirm the efficacy of both Z-FA-FMK and E64d in SMA mice *in vivo*, providing novel targets and approaches to treating SMA.

Since E64d can pass BBB and is easy to be administered to neonatal mice, this drug was injected into SMA mice daily (every day starting from day 1). This may be why E64d showed a stronger effect than Z-FA-FMK in long-term and significantly increased in the life span of SMA mice. Nevertheless, Z-FA-FMK which is only treated for 3 days, showed a trend in increasing the life span of SMA mice. Both Z-FA-FMK and E64d significantly increase the proportion of spinal motor neurons in spinal cord section at day 6 in SMA mice (**Fig. 10**). **These data together demonstrate the effectiveness of candidate compounds *in vivo* and suggest a novel approach to treat SMA through targeting cysteine proteases.** E64d is a potent inhibitor of thiol protease including CTSB (Hook et al., 2015; Inubushi et al., 1994; Romine et al., 2017; Trinchese et al., 2008; Tsubokawa et al., 2006). E64d has also proved to be safe to humans as shown in a clinical study for Alzheimer's disease and traumatic brain injury (this trial was stopped because lack of efficacy but the drug has proven to be safe to human). Given that E64d can significantly increase the proteins levels of SMN, the survival of spinal motor neurons, and the life span of SMA mice *in vivo*, our study not only identifies novel targets for regulating SMN proteins, but also provides a new candidate therapeutic agent for SMA.

3. Novelty and significance of our work versus previous studies

Table 1 summarized previous studies that are related to the susceptibility of SMN- $\Delta 7$ proteins and the upregulation of SMN proteins for SMA. As we can see, though previous studies have shown that SMN- $\Delta 7$ proteins are susceptible to degradation, they are focused on proteasome pathway, phosphorylation, and autophagy. The only study showed the role of cysteine protease in the SMN degeneration is from Strayer's group, in which only Calpain 1 was studied. Furthermore, the protective effects against motor neuron degeneration by cysteine protease inhibitors have not been examined before. During the review of our manuscript, a study reported and examined the role of Calpain 1 in SMA models (de la

Fuente et al., 2018) , but did not examine any Cathepsin (lysosomal-dependent proteases) ; we have included a discussion of this study in the Discussion section of the revised manuscript.

Our work is novel and significant in the following aspects: 1) first work to show that both non-lysosomal (e.g. CAPN1/7) and lysosomal cysteine proteinases (e.g. CTSL/CTSB) mediate the degradation of SMN proteins; 2) demonstrate the novel effects of Z-FA-FMK and E64d, two cysteine protease inhibitors, in increasing SMN proteins in SMA models both *in vitro* and *in vivo*; 3) reveal the efficacy of Z-FA-FMK and E64d in rescuing motor neuron loss in iPSC-based culture model and SMA mouse model; E64d also significantly increases the life span of SMA mice. Together, our work demonstrates the novel role of cysteine proteases (both non-lysosomal and lysosomal cysteine proteases) in regulating SMN protein stability, and reveal novel targets and candidate for rescuing motor neuron degeneration for the treatment of SMA.

Point-by-point response to reviewers' remaining questions:

Reviewer 1:

1. Evaluation of the HTS assay system is insufficient. Positive control is lacking and information of the HTS method is poor.

- Thank you for pointing this out. We have now included a detailed description about the drug screening method (in the Method section) and the examination of positive controls (**Fig. S1**).

2. The authors described that Z-FA-FMK could elongate the life span of SMA model mice in the results section. However, Z-FA-FMK did not show any positive effect on mouse survival with statistical significance. Furthermore, it is unclear why E64d, which was not a hit compound, was selected in the next *in vivo* experiments, even though there were several cysteine protease inhibitors.

- We have now modified our description in the text to “Z-FA-FMK showed a trend in increasing the life span of SMA mice”. Since Z-FA-FMK could not pass BBB, this compound was administrated to SMA mice only for a short period of time. E64d was selected because this drug can pass BBB and was proved to be safe from a previous clinical trial. Indeed, E64d significantly mitigated motor neuron degeneration and increased the life span of SMA mice (see our “*Address to the major concern #2*” for more details about the protective effects of E64d). We have added the related information in the revised manuscript.

3. Details of the method for *in vivo* experiments are lacking. The authors should describe how they injected Z-FA-FMK into lateral cerebral ventricles of postnatal day 1 - day 3 mice, and also how they decided the dose of compounds. It is unclear whether the amount of Z-FA-FMK 60 ng (155 microM) of 1 microL per day is appropriate for the treatment.

- These details are now added in the revised manuscript (Method section and Results section). The concentration of Z-FA-FMK used in ICV injection is based on the amount in another paper (Hara et al., 1997), which is also cited in the revised paper.

4. The authors should investigate the SMN2 protein levels in spinal cord of SMN model mice after treatment with Z-FA-FMK and E64d to clarify the POC *in vivo*.

- Thanks for this great suggestion. We have examined the proteins in SMA mice and found that both Z-FA-FMK and E64d can significantly increase the protein levels of SMN proteins (**Fig. 10A, G**).
5. (*Minor points 1*) The quality of all images to show altered GFP signals is poor with a high background noise. The authors had better use the Luc reporter system.
- GFP signals represent SMN- Δ 7 proteins which will degrade fast; therefore the GFP signals are weak in cells. When measuring the GFP fluorescence by *ImageJ* software, we always consider and calculate the background intensity and remove it from the cellular signals of each well. In the future, we will replace GFP with luciferase for further improvement of the system.
6. (*Minor points 2*) The authors should add graphs to show the screening results by plotting altered ratios compared with DMSO control for each compound.
- We have now included a graph to show the screening results which identified 14 compounds that could brighten the GFP fluorescence by more than 0.5-fold compared to that of DMSO-treated cells (**Fig. 2H**).
7. (*Minor points 3*) In Figure 2H, Western blotting band is not appropriate. GAPDH band is also altered by the addition of compounds, including compound #8 up to 2-fold change. The authors should also add the calculated data to show the changes.
- GAPDH alteration is attributed to the bias of loading samples, rather than to the addition of compounds. We performed the Western blot again and quantified the data (n=4), as shown in Figure 3 (the relative levels of SMN proteins compared to GAPDH proteins were utilized for comparisons, which eliminate the variations caused by loading different amount of samples). From the quantification data, compound #8 is the most effective hit to increase endogenous SMN proteins in type I SMA fibroblast cells.
8. (*Minor points 4*) In Figure 4A, characterization of differentiated motor neurons is lacking.
- This protocol is based on our well-established protocol. We have previously examined and characterized the motor neuron differentiation from these SMA iPSCs (Xu et al., 2016). We have now described this work in more detail and added detailed information.
9. (*Minor points 5*) In Figure 4B, the band of actin is also altered when adding Z-FA-FMK.
- To accurately quantify the protein levels, actin was used as a loading control here. When comparing between different groups, the levels of SMN proteins relative to actin proteins were compared between different groups to eliminate the variations caused by the different loading amounts of samples. We also statistically analyzed the quantification data between different groups and observed a dose-dependent increase of SMN proteins by Z-FA-FMK.

Reviewer 2:

1. For most of the quantification data shown, the author indicated that mean \pm s.e.m. was shown. The information about replicate/n number for those quantifications should be provided.
 - Thank you for bringing this up. We have now included this information in the figure legends.

2. In Figure 4A, the time point for each stage should also be clearly stated either in the figure, or in the legend.

- We have now added this information in the figure legends.

3. In Figure 7, the authors identified 3 cysteine proteases CAPN1, CAPN7, CTSL as important to regulate SMN protein stability. However, the effects of Z-FA-FMK on these specific proteases were not examined or discussed.

- Thank you for this great suggestion. As we discussed in the above section (*Address to the major concerns #1*), we have examined the role of cysteine proteases from multiple angles and identified that both lysosomal and non-lysosomal proteases mediate the degradation of SMN proteins. We then examined the effects of Z-FA-FMK on these proteases and observed an inhibition of SMN protein degradation by Z-FA-FMK after overexpressing cysteine proteases (**Fig. S7**).

4. In Figure 8A, this figure is difficult to understand. The author should clearly explain the results.

- In Figure 8A, these images represent the position (X-axis) versus timing (Y-axis) kymographs. X-axis in the kymograph represents the positions along neuronal axon, Y-axis represents the duration that we detected mitochondrial movement (i.e. 5 minutes). Each line in the image represents a mitochondrion; if this mitochondrion does not move during 5-minute imaging, it will not change its position along X-axis, which will lead to a vertical line. From these representative images, we could see less vertical lines in "SMA+Z-FA-FMK" group compared to "SMA+DMSO" group, suggesting an increase of mitochondrial transport by Z-FA-FMK. A detailed explanation has been added.

5. In Figure 8 and 9, mitochondria transport and neuron degeneration were examined to confirm the functional relevance of SMN protein stabilization mediated by Z-FA-FMK. Ideally, neuronal activities should be examined to eventually confirm the functional relevance, or the author should discuss about this.

- We have now discussed the future examination of neuronal activities in the Discussion section.

6. In Figure 9A and B, although there was a decrease in ISI1+ motor neurons in SMA+DMSO group, the proportion of DAPI+ nuclei looks similar (i.e. similar numbers of total cells were still growing in each condition). How do the authors address the effects of differentiation variations/ motor neuron purity among different groups in their quantification?

- SMA is characterized by the specific degeneration of spinal motor neurons. As we reported before, the initial specification of spinal motor neuron lineage was not affected in SMA group. So the initial motor neuron proportion is similar between control and SMA groups (the efficient generation of Olig2+ motor neuron progenitors were observed in all groups). To ensure a rigorous comparison, multiple regions were selected blindly from triplicate samples as we described before. The degeneration of spinal motor neurons happened in long-term cultures (e.g. changes in morphology and accumulations of swellings), which is further supported by the

increased apoptosis in the long-term cultures (**Fig. 4E**). We have now described these more in detail in the revised manuscript.

7. In Figure 9C, the author may also enlarge part of the figure to better display the axon swelling and breakdown.

- An enlarged part was included to show the axon swellings and breakdown.

8. In Figure 10C and D, there is no WT control. Besides, if there is no DAPI staining for total nuclei, how did the author quantify the percentage of positive cells (Figure 10D)?

- Thank you for pointing this out. The WT group would be the same as shown for Z-FA-FMK treatment and here we focused on comparing vehicle- and E64d- treated groups. Yes, nuclei staining (Hoechst) was performed; it was not included before for showing other two staining more clear. We have now also included Hoechst in the images.

Reviewer 3:

1. The susceptibility of SMN Δ 7 to proteases has been previously elucidated and the potential of inhibitors of this process for SMA have been previously proposed by studies from Burnett/Fishbeck, Dreyfuss, and Rubin laboratories.

- Though the susceptibility of SMN Δ 7 to proteases has been previously elucidated, previous studies have been focused on ubiquitin proteasome and Calpain 1 (please see the summarized **Table 1**). In addition to Calpain 1, our data identified the novel role of both non-lysosomal (e.g. Calpain 7) and lysosomal cysteine proteases (e.g. CTSB/CTSL) in degrading SMN proteins (**Fig. 7; Fig. S7**). This is of high importance because these members are different from Calpain 1 and have unique roles under pathological conditions. Please see the above section (*Address to major concerns of the reviewers*) for detailed information on the previous studies and our findings.

2. Protease inhibitors, such as Z-FA-FMK, are likely to have pleiotropic effects rather than selective effect on SMN Δ 7. It is possible, and whatever effects Z-FA-FMK has may be on these specific proteases unrelated to the SMN increase and will be deleterious to cells or an organism in longer term experiments.

- Thanks for bringing up this great point. One of candidate compounds, E64d, has been used in a clinical trial for Alzheimer's disease and traumatic brain injury (from *Medtrack*). The drug has been approved to be safe to humans (this trial was stopped because lack of efficacy). In our study, E64d can significantly increase the proteins levels of SMN, the survival of spinal motor neurons, and the life span of SMA mice in vivo, suggesting that E64d serves a new candidate therapeutic agent for SMA. We have included a discussion on this issue in the revised manuscript.

References Cited:

- Abera, M.B., Xiao, J., Nofziger, J., Titus, S., Southall, N., Wei, Z., Moritz, K.E., Ferrer, M., Cherry, J.J., and Androphy, E.J. (2016). ML372 blocks SMN ubiquitination and improves spinal muscular atrophy pathology in mice. *Jci Insight* 1, e88427.
- Burnett, B.G., Muñoz, E., Tandon, A., Kwon, D.Y., Sumner, C.J., and Fischbeck, K.H. (2009). Regulation of SMN Protein Stability. *Molecular and Cellular Biology* 29, 1107-1115.
- Cherry, J.J., Osman, E.Y., Evans, M.C., Choi, S., Xing, X.C., Cuny, G.D., Glicksman, M.A., Lorson, C.L., and Androphy, E.J. (2013). Enhancement of SMN protein levels in a mouse model of spinal muscular atrophy using novel drug-like compounds. *Embo Mol Med* 5, 1103-1118.
- de la Fuente, S., Sansa, A., Periyakarupiah, A., Garcera, A., and Soler, R.M. (2018). Calpain Inhibition Increases SMN Protein in Spinal Cord Motoneurons and Ameliorates the Spinal Muscular Atrophy Phenotype in Mice. *Mol Neurobiol*.
- Ebert, A.D., Yu, J., Rose, F.F., Jr., Mattis, V.B., Lorson, C.L., Thomson, J.A., and Svendsen, C.N. (2009). Induced pluripotent stem cells from a spinal muscular atrophy patient. *Nature* 457, 277-280.
- Fuentes, J.L., Strayer, M.S., and Matera, A.G. (2010). Molecular Determinants of Survival Motor Neuron (SMN) Protein Cleavage by the Calcium-Activated Protease, Calpain. *Plos One* 5.
- Hara, H., Friedlander, R.M., Gagliardini, V., Ayata, C., Fink, K., Huang, Z., Shimizu-Sasamata, M., Yuan, J., and Moskowitz, M.A. (1997). Inhibition of interleukin 1beta converting enzyme family proteases reduces ischemic and excitotoxic neuronal damage. *Proc Natl Acad Sci U S A* 94, 2007-2012.
- Heier, C.R., and DiDonato, C.J. (2009). Translational readthrough by the aminoglycoside geneticin (G418) modulates SMN stability in vitro and improves motor function in SMA mice in vivo. *Hum Mol Genet* 18, 1310-1322.
- Hook, G., Jacobsen, J.S., Grabstein, K., Kindy, M., and Hook, V. (2015). Cathepsin B is a New Drug Target for Traumatic Brain injury Therapeutics: evidence for E64d as a Promising Lead Drug Candidate. *Front Neurol* 6.
- Inubushi, T., Kakegawa, H., Kishino, Y., and Katunuma, N. (1994). Specific Assay-Method for the Activities of Cathepsin L-Type Cysteine Proteinases. *J Biochem-Tokyo* 116, 282-284.
- Makhortova, N.R., Hayhurst, M., Cerqueira, A., Sinor-Anderson, A.D., Zhao, W.N., Heiser, P.W., Arvanites, A.C., Davidow, L.S., Waldon, Z.O., and Steen, J.A. (2011). A screen for regulators of survival of motor neuron protein levels. *Nature Chemical Biology* 7, 544-552.
- Rietz, A., Li, H., Quist, K.M., Cherry, J.J., Lorson, C.L., Burnett, B., Kern, N.L., Calder, A.N., Fritsche, M., and Lusic, H. (2017). Discovery of a Small Molecule Probe that Post-translationally Stabilizes the Survival Motor Neuron Protein for the Treatment of Spinal Muscular Atrophy. *Journal of Medicinal Chemistry* 60, 4594.
- Rodriguezmuela, N., Parkhitko, A., Grass, T., Gibbs, R.M., Norabuena, E.M., Perrimon, N., Singh, R., and Rubin, L.L. (2018). Blocking p62/SQSTM1-dependent SMN degradation ameliorates Spinal Muscular Atrophy disease phenotypes. *Journal of Clinical Investigation*.
- Romine, H., Rentschler, K.M., Smith, K., Edwards, A., Colvin, C., Farizatto, K., Pait, M.C., Butler, D., and Bahr, B.A. (2017). Potential Alzheimer's Disease Therapeutics Among Weak Cysteine Protease Inhibitors Exhibit Mechanistic Differences Regarding Extent of Cathepsin B Up-Regulation and Ability to Block Calpain. *Eur Sci J* 13, 38-59.
- S, C., and G, D. (2010). A degron created by SMN2 exon 7 skipping is a principal contributor to spinal muscular atrophy severity. *Genes Dev* 24, 438-442.
- Trinchese, F., Fa, M., Liu, S., Zhang, H., Hidalgo, A., Schmidt, S.D., Yamaguchi, H., Yoshii, N., Mathews, P.M., Nixon, R.A., *et al.* (2008). Inhibition of calpains improves memory and synaptic transmission in a mouse model of Alzheimer disease. *J Clin Invest* 118, 2796-2807.
- Tsubokawa, T., Solaroglu, I., Yatsushige, H., Cahill, J., Yata, K., and Zhang, J.H. (2006). Cathepsin and calpain inhibitor E64d attenuates matrix metalloproteinase-9 activity after focal cerebral ischemia in rats. *Stroke* 37, 1888-1894.

Wolstencroft, E.C., Mattis, V., Bajer, A.A., Young, P.J., and Lorson, C.L. (2005). A non-sequence-specific requirement for SMN protein activity: the role of aminoglycosides in inducing elevated SMN protein levels. *Human Molecular Genetics* 14, 1199-1210.

Xu, C.C., Denton, K.R., Wang, Z.B., Zhang, X., and Li, X.J. (2016). Abnormal mitochondrial transport and morphology as early pathological changes in human models of spinal muscular atrophy. *Dis Model Mech* 9, 39-49.

March 5, 2019

RE: Life Science Alliance Manuscript #LSA-2018-00268-TR

Dr. Xue-Jun Li
University of Illinois
Department of Biomedical Sciences
1601 Parkview Avenue
College of Medicine at Rockford
Rockford, IL 61107

Dear Dr. Li,

Thank you for submitting your revised manuscript entitled "Drug screening with human SMN2 reporter identifies SMN protein stabilizers to correct SMA pathology". I appreciate many of the introduced changes, and would like to ask you to still address the below listed points prior to acceptance of your work:

- I appreciate the provided quantifications, however, the effects on SMN stability are often rather subtle. This should get mentioned in the manuscript text.
- please mention the statistical test used in the figure legends
- the pulldown of Myc seems to not have worked well in the SMN2a/CAPN7 negative control experiment, please replace

A. FINAL FILES:

-- Summary blurb (enter in submission system): A short text summarizing in a single sentence the study (max. 200 characters including spaces). This text is used in conjunction with the titles of

papers, hence should be informative and complementary to the title. It should describe the context and significance of the findings for a general readership; it should be written in the present tense and refer to the work in the third person. Author names should not be mentioned.

B. MANUSCRIPT ORGANIZATION AND FORMATTING:

Sincerely,

Andrea Leibfried, PhD
Executive Editor
Life Science Alliance
Meyerohofstr. 1
69117 Heidelberg, Germany
t +49 6221 8891 502
e a.leibfried@life-science-alliance.org
www.life-science-alliance.org

Dr. Xue-Jun Li, Associate Professor
Department of Biomedical Sciences
Regenerative Medicine & Disability Research Lab
College of Medicine at Rockford
Department of Bioengineering
University of Illinois at Chicago
Phone: 815-395-5882
Email: xjli23@uic.edu

March 10, 2019

Dear Dr. Leibfried,

Thank you for your positive feedbacks and helpful suggestions. Also thank you for inviting us to submit the final version in which we have addressed all the listed points (labelled in blue color).

Response to the listed points:

1. I appreciate the provided quantifications; however, the effects on SMN stability are often rather subtle. This should get mentioned in the manuscript text.
 - We appreciate this great suggestion. Though the increase of SMN- $\Delta 7$ by targeting individual cysteine protease is only 5-20%, the total increase of SMN (both SMN-FL and SMN- $\Delta 7$) through inhibiting the activity of multiple proteases would be higher. For SMA, Type I patients die before the age of 2, while type II patients who have 50% higher levels of functional SMN could live well into adulthood. Therefore, a moderate increase in SMN levels would significantly improve the survival of motor neurons. Moreover, compounds that increase SMN stability can be combined with drugs targeting other mechanisms (e.g. increasing SMN expression) in the future for combination therapy. We have now included a discussion on this issue in the revised manuscript.
2. Please mention the statistical test used in the figure legends.
 - The related information has been added to the figure legends.
3. The pulldown of Myc seems to not have worked well in the SMN2a/CAPN7 negative control experiment, please replace.
 - Thank you for pointing this out. We have replaced this in the revised manuscript.

Sincerely,

Xue-Jun Li

March 12, 2019

RE: Life Science Alliance Manuscript #LSA-2018-00268-TRR

Dr. Xue-Jun Li
University of Illinois
Department of Biomedical Sciences
1601 Parkview Avenue, Room E401
College of Medicine at Rockford
Rockford, IL 61107

Dear Dr. Li,

Thank you for submitting your Research Article entitled "Drug screening with human SMN2 reporter identifies SMN protein stabilizers to correct SMA pathology". I appreciate the introduced changes and it is a pleasure to let you know that your manuscript is now accepted for publication in Life Science Alliance. Congratulations on this interesting work.

DISTRIBUTION OF MATERIALS:

Again, congratulations on a very nice paper. I hope you found the review process to be constructive and are pleased with how the manuscript was handled editorially. We look forward to future exciting submissions from your lab.

Sincerely,

Andrea Leibfried, PhD
Executive Editor
Life Science Alliance
Meyershofstr. 1
69117 Heidelberg, Germany
t +49 6221 8891 502
e a.leibfried@life-science-alliance.org
www.life-science-alliance.org